# Sex differences in cerebellar synaptic transmission and sex-specific responses to autism-linked *Gabrb3* mutations in mice

Audrey A Mercer[1,2]*, Kristin J Palarz[2,3], Nino Tabatadze[2], Catherine S Woolley[1,2], Indira M Raman[1,2,3]*

[1]Northwestern University Interdepartmental Neuroscience Program, Northwestern University, Evanston, United States; [2]Department of Neurobiology, Northwestern University, Evanston, United States; [3]Integrated Science Program, Northwestern University, Evanston, United States

**Abstract** Neurons of the cerebellar nuclei (CbN) transmit cerebellar signals to premotor areas. The cerebellum expresses several autism-linked genes, including *GABRB3*, which encodes $GABA_A$ receptor β3 subunits and is among the maternal alleles deleted in Angelman syndrome. We tested how this *Gabrb3* m-/p+ mutation affects CbN physiology in mice, separating responses of males and females. Wild-type mice showed sex differences in synaptic excitation, inhibition, and intrinsic properties. Relative to females, CbN cells of males had smaller synaptically evoked mGluR1/5-dependent currents, slower Purkinje-mediated IPSCs, and lower spontaneous firing rates, but rotarod performances were indistinguishable. In mutant CbN cells, IPSC kinetics were unchanged, but mutant males, unlike females, showed enlarged mGluR1/5 responses and accelerated spontaneous firing. These changes appear compensatory, since mutant males but not females performed indistinguishably from wild-type siblings on the rotarod task. Thus, sex differences in cerebellar physiology produce similar behavioral output, but provide distinct baselines for responses to mutations.

*For correspondence: audrey.a.
mercer@gmail.com (AAM); i-
raman@northwestern.edu (IMR)

**Competing interest:** See
page 21

**Reviewing editor:** Michael
Häusser, University College
London, United Kingdom

## Introduction

Neurons in the cerebellar nuclei (CbN) form the final stage of cerebellar processing. They integrate synaptic inhibition from Purkinje neurons of the cerebellar cortex with synaptic excitation from mossy fibers and inferior olivary fibers, thereby generating the sole output of the non-vestibulocerebellum. Consequently, disruptions in intrinsic or synaptic signaling anywhere in the cerebellar circuit are likely to manifest themselves in CbN cell activity, either as altered or compensatory responses. Recent work has drawn attention to the fact that many genes disrupted in autism spectrum disorders (ASD) are expressed in the cerebellum (*Fatemi et al., 2012*; *Wang et al., 2014*), suggesting that, regardless of the etiology of the phenotypes associated with the condition, cerebellar processing may be affected. Indeed, cerebellar abnormalities are consistently found in post-mortem examinations of autistic brains, which reveal decreases in size and number of Purkinje and/or CbN cells (*Bauman and Kemper, 1985*; *Arin et al., 1991*; *Bailey et al., 1998*; *Kemper and Bauman, 1998*; *Whitney et al., 2008*). Mice with the 15q11-13 duplication, a model for one form of autism, show defects in cerebellar learning (*Piochon et al., 2014*), and mutation of *Tsc1* in Purkinje cells alone can recapitulate several autism-like behaviors in mice (*Tsai et al., 2012*). Studying the cerebellum in mouse models of autism may therefore yield insight into how cerebellar output is generated and to what extent cerebellar processing is plastic or disrupted in the face of genetic abnormalities.

**eLife digest** The cerebellum is a part of the brain that plays a role in controlling movement, coordination and balance. It also contributes to cognitive processes that do not relate to movement. Changes in the cerebellum are often seen in individuals with autism spectrum disorders, and many genes implicated in autism are active in the cerebellum. One of these genes encodes part of a receptor for the signaling molecule GABA, and is called *GABRB3* in humans. Each person usually inherits one copy of the *GABRB3* gene from each parent. However, if a mother does not pass on this gene, her child may develop Angelman syndrome. Those affected by this disorder show impaired movement and cognitive abilities, and often show signs of autism.

To explore how neural signals in the cerebellum might change in Angelman syndrome, Mercer et al. compared mice that lack a copy of the mouse equivalent of the gene (called *Gabrb3*) from their mothers to "control" mice with two intact copies of the gene. The experiments unexpectedly revealed that key brain cells in the cerebellum of male control mice were different from the same cells in female mice. The cells in males had lower baseline levels of electrical activity and responded differently to signals from other cells. The differences arose partly because a group of receptors – called metabotropic glutamate receptors – were more easily activated in the brain cells in females than in males. Nevertheless, both male mice and female mice did equally well at learning to balance on a rotating rod, which is a skill that is controlled by the cerebellum. In other words, the cerebellum works differently in male and female mice but produces the same output.

Mutant male mice performed just as well as non-mutant, control males at learning to balance on the rotating rod. By contrast, female mutants did not improve during training on the same task. Measuring the activity of cells in the cerebellum showed that the metabotropic glutamate receptors in cells from mutant male mice had changed so that they responded more like those of females. However, the responses of mutant female mice did not change compared to control female mice. This result suggests that the changes in the brain cells of male mutant mice helped compensate for the *Gabrb3* mutation. It also shows that baseline differences in the brains of male and female animals can make them respond differently to mutations associated with genetic disorders.

*GABRB3,* which encodes the β3 subunit of the GABA$_A$ receptor (GABA$_A$R), is an autism-linked gene with cerebellar expression. It is among the genes affected in Angelman syndrome, a condition in which patients show developmental delay, motor stereotypy, and movement disorders, among other symptoms (*Williams et al., 2006*). These patients have a microdeletion of the 15q11-13 region of the maternal chromosome, which spans multiple genes (*Knoll et al., 1989*), of which *UBE3A* and *GABRB3* have been the most extensively studied. Mice lacking maternal *Ube3a* recapitulate many but not all phenotypes expected for an animal model of Angelman syndrome (*Jiang et al., 1998*; *Allensworth et al., 2011*). Mice lacking only the maternal copy of *Gabrb3* (m-/p+), however, also show many of these phenotypes. Regarding cerebellar function, adult *Gabrb3* m-/p+ mice display deficits in motor learning as measured by the accelerating rotarod task (*DeLorey et al., 2011*). In the cerebellum, β3 is expressed in Purkinje and granule cells, with low expression in the cerebellar nuclei (*Laurie et al., 1992*; *Fritschy and Mohler, 1995*; *Hörtnagl et al., 2013*). Since β3 slows the kinetics of GABA$_A$R currents in expression systems (*Hinkle and Macdonald, 2003*), IPSC kinetics of neurons that express GABA$_A$R β3 are predicted to be accelerated in *Gabrb3* m-/p+ mice, which should disinhibit those cells and alter signaling through the cerebellum. Since all such changes must ultimately be funneled through the cerebellar nuclei, we examined synaptic properties of large, likely glutamatergic, premotor CbN neurons in cerebellar slices from wild-type and *Gabrb3* m-/p+ mice.

Because of a reported sex difference in motor behavior of m-/p+ mice (*DeLorey et al., 2011*), we also compared electrophysiological responses in males and females. These comparisons revealed that, even in wild-type mice, a wide range of basal synaptic and intrinsic properties of CbN cells differ between the sexes. Males and females also respond differently to the *Gabrb3* m-/p+ mutation, which triggers an augmentation of mGluR1/5 responses and intrinsic firing rates in male but not female mutant mice, which is apparently compensatory on a motor learning task.

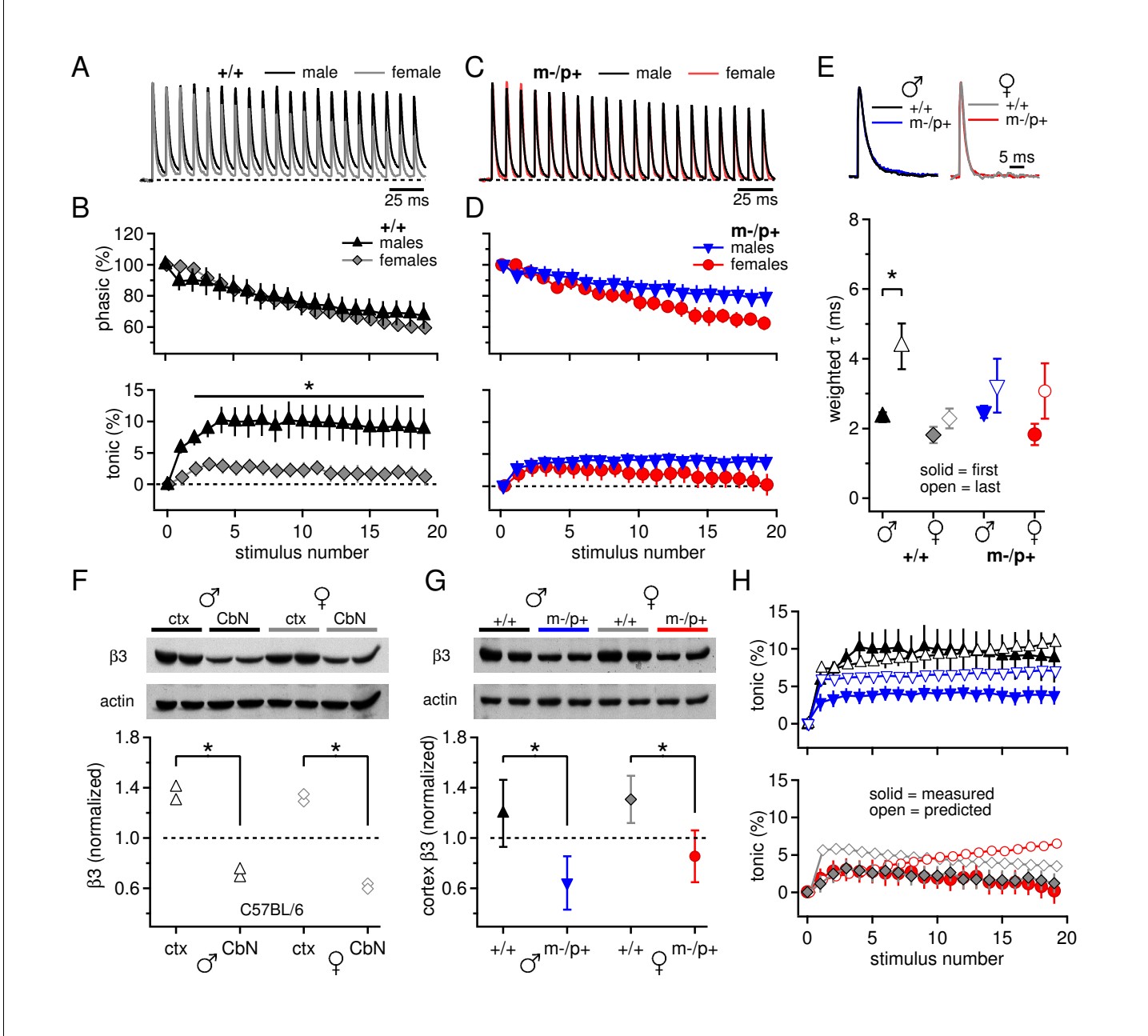

**Figure 1.** Sex differences in CbN synaptic currents and sex-specific responses to the *Gabrb3* m-/p+ mutation. (A) 100-Hz trains of synaptic currents evoked in CbN cells from male and female wild-type mice, normalized to the first peak. Dotted line, baseline holding current. (B) Mean amplitudes of phasic (upper panel) and tonic (lower panel) synaptic currents as a percentage of the first peak synaptic current vs. stimulus number. Dotted line, 0 current. (C,D) As in A, B, but for cells from male and female m-/p+ mice. (E) Top: example IPSCs evoked by a single stimulus, normalized to the peak current. Bottom: Solid symbols: weighted $\tau_{decay}$ for IPSCs from a single stimulus. Open symbols: weighted $\tau_{decay}$ for the last IPSC in the train. (F) Representative blot (top) and quantification (bottom) for β3 subunit expression in the cerebellar cortex *vs.* the cerebellar nuclei in C57BL/6 mice. Each symbol represents the normalized value for one lane. (G) Representative blot (top) and quantification (bottom) for normalized β3 subunit protein expression in the cerebellar cortex of *Gabrb3* mice. (H) Solid symbols: measured tonic current, re-plotted from (B) and (D). Open symbols: predicted tonic current, calculated from the weighted $\tau_{decay}$ of the first and last IPSCs. Symbol color code as in (E). In all figures, data are plotted as mean ± SEM. Asterisks indicates statistically significant differences.

The following figure supplements are available for figure 1:

**Figure supplement 1.** Sex differences in tonic current and IPSC decay kinetics in C57BL/6 mice.

*Figure 1 continued on next page*

*Figure 1 continued*

**Figure supplement 2.** Tonic synaptic current, likely mediated by mGluR1/5, may be modulated by environmental conditions such as chronic stress.

## Results

Since expression of the β3 subunit is expected to be reduced in *Gabrb3* m-/p+ mice, we first tested for direct effects of the mutation on synaptically evoked GABA$_A$R responses. Male and female *Gabrb3* m-/p+ mice and littermate controls were considered separately because of sex differences reported in behavioral responses to the mutation (*DeLorey et al., 2011*). Whole-cell recordings were made from large CbN cells in cerebellar slices from P17-P24 mice, and voltage-clamped synaptic currents were evoked with 100-Hz, 200-ms trains of electrical stimuli to evoke Purkinje cell-mediated IPSCs (*Telgkamp and Raman, 2002*). Because these stimulus trains necessarily also elicit glutamate release from excitatory mossy fibers, AMPA and NMDA receptors were blocked by 5 µM DNQX and 10 µM CPP. The trains evoked outward synaptic currents dominated by GABA$_A$R-mediated current from Purkinje cell stimulation (*Telgkamp and Raman, 2002*; *Figure 1A*). During trains, the synaptic current did not decay fully, so that with the onset of the next stimulus, the additional 'phasic' current, i.e., synaptic current evoked by release elicited by a single stimulation, summed with the preceding 'tonic' current, i.e., residual or accumulating synaptic current from previous release events. The tonic component is of interest because it is largely responsible for suppressing intrinsic firing by CbN cells (*Telgkamp and Raman, 2002*; *Person and Raman, 2012*). In wild-type males and females, the phasic components largely overlaid one another (*Figure 1A,B*; repeated-measures ANOVA p=*0.78*). In contrast, the synaptic current decayed to a greater extent in females, leading to a tonic component that was more than 3-fold larger in wild-type males than females (*Figure 1A,B*, +/+ males n=11, +/+ females n=10, p=*0.005*). This result was replicated in C57BL/6 males and females (*Figure 1—figure supplement 1A,B*), demonstrating that elements of basal synaptic transmission in the CbN of mice differ between the sexes, even in pre-pubertal animals.

When the experiment was repeated in m-/p+ mice, mutant males showed altered synaptic responses, but mutant females did not, such that both sexes had small tonic currents that resembled those of wild-type females (*Figure 1C,D*). Consequently, cells from mutant males had a smaller tonic component than wild-type males (m-/p+ males n=11, p=*0.029*), but wild-type and mutant females did not differ (m-/p+ females n=11, p=*0.94*). Thus, the *Gabrb3* m-/p+ mutation leads to a change in tonic synaptic current in males but not in females, eliminating the sex difference present in wild-type animals.

To test whether the differences in tonic current arose directly from differences in GABA$_A$R kinetics, we measured the decay kinetics of synaptic currents evoked by single stimuli. The weighted $\tau_{decay}$ was 2.3 ± 0.1 ms in wild-type males (n=22) and 1.8 ± 0.2 ms in wild-type females (n=9, p=*0.07*). Similar values were seen in cells from C57BL/6 mice, and combining the datasets demonstrated that cells from females indeed had significantly slower IPSC decay times than males (*Figure 1—figure supplement 1C*, males 2.4 ± 0.1, n=33; females 1.8 ± 0.2, n=21; p=*0.008*). Despite the fact that GABA$_A$R β3 expression is expected to slow the deactivation of GABA$_A$R currents (*Hinkle and MacDonald 2003*), the mean time constants in cells from m-/p+ mice were indistinguishable from those of wild-type, for both males and females (*Figure 1E*, solid symbols; m-/p+ males 2.5 ± 0.2 ms, n=16, p=*0.5 vs. +/+*; m-/p+ females 1.8 ± 0.2 ms, n=6, p=*1.0 vs. +/+*). This similarity suggests that the kinetics of evoked synaptic currents in large CbN cells do not depend strongly on GABA$_A$R β3, possibly because *Gabrb3* m-/p+ does not actually result in reduced cerebellar expression, or because the GABA$_A$R β3 subunit is not strongly expressed in the cerebellar nuclei even in wild-type mice.

We therefore measured β3 subunit expression in CbN tissue in C57BL/6 male and female mice, and compared it to β3 subunit expression in the cerebellar cortex. Indeed, β3 subunit expression was relatively low in the CbN; β3 expression in the cerebellar cortex was about twice that in the cerebellar nuclei, in both males and females (*Figure 1F*, males, n=2, p=*0.02*, females, n=2, p=*0.004*). These data corroborate previous results from immunostaining and in situ hybridization that report cerebellar GABA$_A$R β3 expression to be primarily in granule cells and Purkinje cells (*Laurie et al.,*

1992; *Fritschy and Mohler, 1995*; *Hörtnagl et al., 2013*). Next, to test whether GABA$_A$R β3 levels are detectably reduced in *Gabrb3* m-/p+ mice, we compared expression in the cerebellar cortex across all four groups of mutants and wild-type siblings of both sexes. Consistent with a gene dosage effect, loss of the maternal *Gabrb3* allele significantly reduced GABA$_A$R β3 expression to about half in both sexes (*Figure 1G*, males, n=3, p=0.001; females, n=3, p=0.005). In wild-type animals, expression in the cerebellar cortex was indistinguishable in males and females, in both *Gabrb3* sibling controls (n=3, p=0.21) and in C57BL/6 mice (n=6, p=0.37). GABA$_A$R β3 expression in the cerebellar nuclei was also significantly reduced in mutant males and females (p≤0.001, both sexes). Thus, the lack of changes in IPSC decay times in CbN cells of *Gabrb3* m-/p+ mice is more likely to result from sparse GABA$_A$R β3 expression in wild-type mice, rather than a failure of the mutation to reduce expression.

Moreover, the similarity of IPSC decay times for sex-matched wild-type and mutant CbN cells suggests that factors other than GABA$_A$R decay kinetics must account for differences in tonic currents between wild-type and mutant males. To test the extent to which IPSC decay times were

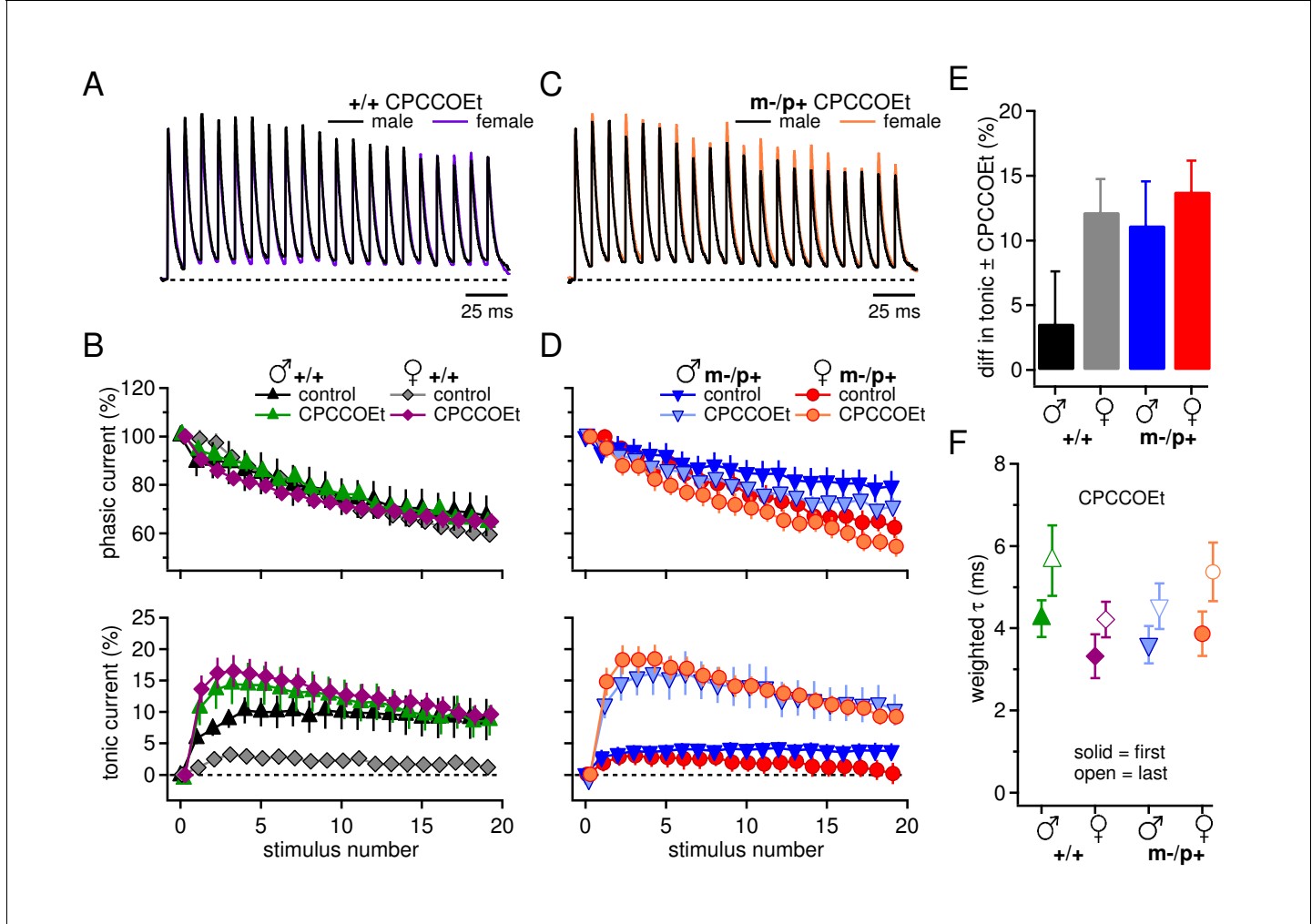

**Figure 2.** Group I mGluRs account for differences in synaptic currents and responses to the *Gabrb3* m-/p+ mutation. (A) Synaptic currents in the presence of CPCCOEt evoked in CbN cells from male and female wild-type mice. Dotted line, baseline holding current. (B) Mean amplitudes of phasic (upper panel) and tonic (lower panel) synaptic currents recorded in CPCCOEt as a percentage of the peak current evoked by the first stimulus in each train *vs.* stimulus number. Control data from *Figure 1* are superimposed for comparison. Dotted line, 0% current. (C, D) As in B, but for cells from male and female m-/p+ mice. (E) Percent difference in tonic current ± CPCCOEt for each group, calculated from the difference between the mean tonic current for stimuli 5–10 in control and CPCCOEt solutions. (F) Solid symbols: weighted τ$_{decay}$ for each group in CPCCOEt. Open symbols: weighted τ$_{decay}$ from the last IPSC of the train, in CPCCOEt.

predictive of tonic currents, we calculated the expected tonic current based on IPSC decay times and compared it to the measured tonic current. The decay time constants, however, lengthened during the train, such that the last IPSC was significantly longer than the first for wild-type males (*Figure 1E*, open symbols; weighted $\tau_{decay}$ for last IPSC, +/+ males, 4.4 ± 0.7 ms, n=10, p=*0.013 vs.* 1st IPSC; m-/p+ males, 3.2 ± 0.8 ms, n=11, p=*0.36*; +/+ females, 2.3 ± 0.3 ms, n=10, p=*0.21*; m-/p+ females, 3.1 ± 0.8 ms, n=11, p=*0.17*). Although the basis for this change is unknown, it is consistent with a decreasing synchrony of neurotransmitter release time as the train progresses, e.g., associated with changing action potential waveforms (*Bischofberger et al., 2002*). We therefore incorporated this gradual change in $\tau_{decay}$ into our calculations (*Materials and Methods*) and estimated the tonic current for each group. As shown in *Figure 1H*, for wild-type males, the predicted and measured currents were closely matched (*solid and open black triangles*), indicating that IPSC kinetics are sufficient to account for the tonic synaptic current. For mutant males and both female groups, however, the IPSC decay times consistently predicted a tonic current that is smaller than in wild-type males but still *larger* than the measured values (*open vs. solid symbols*), suggesting an additional factor either reducing the total outward tonic current or contributing a tonic inward current.

One possibility arose from the fact that in the experiments of *Figure 1*, only fast excitatory transmission was blocked. Previous work, however, shows that glutamate release from mossy fibers can activate group I metabotropic glutamate receptors (mGluR1/5) in CbN neurons, which in turn can increase tonic *inward* currents through multiple mechanisms (*Zhang and Linden, 2006*; *Zheng and Raman, 2011*). We therefore tested the effect of CPCCOEt (100 µM), which effectively blocks group I mGluRs in CbN cells in a manner that is mimicked by a combination of the selective mGluR1 antagonist JNJ16259685 and mGluR5 antagonist MPEP (*Zheng and Raman, 2011*). When trains of IPSCs were evoked, CPCCOEt increased the net outward tonic current, consistent with activation of group I mGluRs in CbN cells of wild-type and mutant mice of both sexes (+/+ males n=11, females n=11; m-/p+ males n=12, females n=9, p<*0.001*). Importantly, CPCCOEt also eliminated the sex difference between wild-type male and female tonic currents (*Figure 2A,B,* p=*0.93*). Additionally, the tonic currents in mutant cells were indistinguishable from those of wild-type cells (*Figure 2C,D*, males, p=*1.0*; females, p=*0.93*), suggesting that the observed sex difference in wild-type synaptic currents results from a larger mGluR1/5 response in females than in males. The difference in tonic current in control and CPCCOEt (*Figure 2E*) provides a measure of the differential contribution of group I mGluRs. Thus, the *Gabrb3* m-/p+ mutation increases the mGluR1/5 response in males, but not in females.

Phasic current, in contrast, was statistically indistinguishable in CPCCOEt relative to control solutions (*Figure 2D*, p=*0.12*). CPCCOEt, however, did consistently prolong the time course of decay of a single IPSC for all four groups (p<*0.001*, in CPCCOEt, +/+ males n=13, +/+ females n=6, m-/p+ males n=10, m-/p+ females n=6, *Figure 2F*). This effect on even a single stimulus suggests that group I mGluRs may be basally activated in a manner that shortens synaptic currents, e.g., by inhibition of presynaptic Ca currents (*Xu-Friedman and Regehr, 2000*).

Next, we examined the mGluR1/5 current directly. In these experiments synaptic inhibition was blocked by SR95531 (10 µM) and strychnine (10 µM), and NMDA receptors were blocked by CPP (10 µM). After a recording was established, the excitatory amino acid transporter blocker DL-TBOA (50 µM) was added to the bath. TBOA increases glutamate spillover (*Brasnjo and Otis, 2001*; *Huang et al., 2004*) and augments the evoked responses of group I mGluRs in the CbN of ~2-week-old rats (*Zhang and Linden, 2006*). In the present experiments, however, application of TBOA had a striking effect even in the absence of synaptic stimulation: it evoked a large standing current of a few hundred pA at -40 mV, which was largely reversed by group I mGluR antagonists. The remaining current was not further altered by a return to control solutions (*Figure 3A*). Consistent with an action at mGluR1/5, the magnitude of the effect of CPCCOEt (n=10) was indistinguishable from that of combined JNJ16259685 (0.2 µM) and MPEP (40 µM, n=29, p=*0.63*) and the data were pooled. To test whether the effect of TBOA resulted from glutamate released in an action-potential-dependent manner, 1 µM TTX was added to the bath in a subset of experiments. The TBOA-induced current was indeed reversed by TTX (*Figure 3B*, C57BL/6 male mice; control -305 ± 99 pA; TBOA -570 ± 139 pA, p=*0.033 vs.* control; TBOA+TTX -338 ± 43 pA, p=*0.2 vs.* control), and the variance of the standing current was reduced. The data are consistent with the idea that spontaneously firing nearby cells continuously release glutamate. In the presence of TBOA, the released glutamate reaches and activates group I mGluRs on CbN

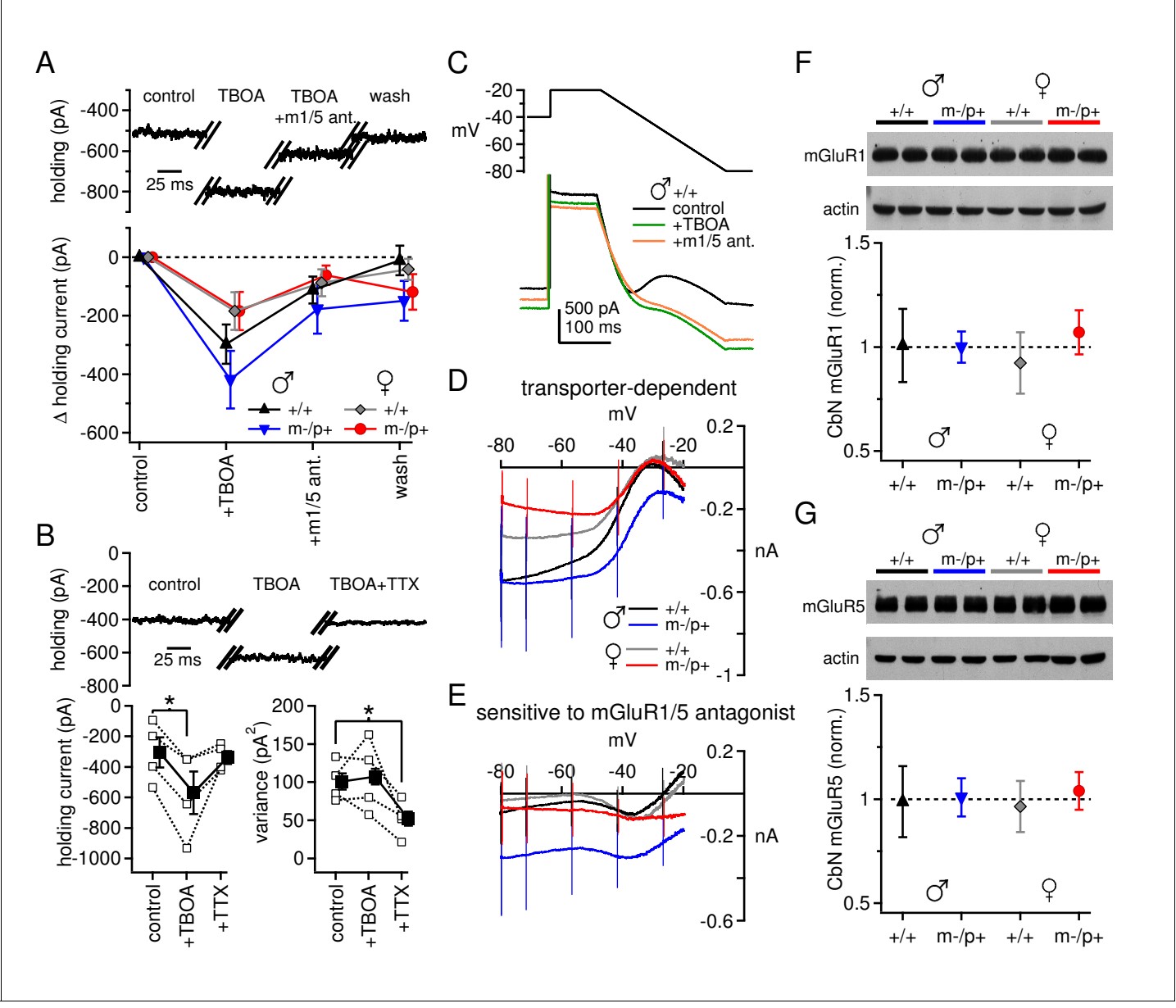

**Figure 3.** Differences in mGluR1/5-dependent currents depend on glutamate access to receptors rather than receptor expression. (**A**) Increase in holding current in a +/+ male mouse by TBOA reversal by group I mGluR antagonists. (**B**) Increase in holding current by TBOA reversal by TTX. Top: sample traces from a C57BL/6 male mouse. Bottom: summary data for current amplitude (left) and variance (right). (**C**) Voltage ramp (top) and current (bottom) for a +/+ male in control, TBOA, and TBOA + mGluR1/5 antagonist(s). Each current is the mean of three traces. (**D**) Transporter-induced current (control current minus current in TBOA) *vs.* voltage. (**E**) TBOA-induced current sensitive to mGluR1/5 antagonist(s) (current in TBOA minus current in TBOA + mGluR1/5 antagonist), *vs.* voltage. In all figures, 'm1/5 ant.' indicates either CPCCOEt or JNJ16259685 + MPEP. (**F**) Representative blot (top) and summary (bottom) for normalized mGluR1 protein expression in the CbN of *Gabrb3* mice. (**G**) As in (**F**), for mGluR5.

cells, which in turn couple to other channels; at this voltage, the inward current is likely carried primarily through L-type Ca channels (*Zheng and Raman, 2011*).

Unlike the evoked currents measured in *Figures 1* and *2*, however, blocking glutamate transport revealed an increase in mGluR1/5-dependent current amplitude at -40 mV that was, on average, *larger* in wild-type males than in wild-type females (*Figure 3A*, +/+ males -297 ± 67 pA, n=9; +/+ females -184 ± 64 pA, n=9, p=0.08). Furthermore, there was no difference between sex-matched mutant *vs.* wild-type groups (m-/p+ males -419 ± 99 pA, n=11, p=0.5 vs. +/+; m-/p+ females -184 ±

65, n=9, p=*0.9 vs. +/+*). These results suggest that the differences in evoked mGluR1/5-dependent currents do not necessarily result from fewer receptors or diminished properties of downstream targets in wild-type males. Instead, the reduced response to evoked release in wild-type males may stem from the accessibility of glutamate to mGluRs when transport is intact, i.e., owing to differences in the efficacy of uptake or proximity of receptors. Moreover, the *Gabrb3* m-/p+ mutation does not apparently alter the maximal mGluR1/5 response, as measured with transporter blockers, in males or females.

To examine the voltage-dependence of the mGluR1/5-dependent current, we applied a voltage ramp to CbN cells from all four groups of *Gabrb3* mice. From an initial holding potential of -40 mV, the voltage was stepped to -20 mV for 100 ms and then ramped down to -80 mV over 250 ms. The ramp revealed a standing inward current that increased in amplitude until about -35 mV, decreased from -35 mV to -50 mV, and then increased monotonically until the end of the ramp (*Figure 3C*). This current profile reflects what has previously been identified as an L-type voltage gated Ca current (*Zheng and Raman, 2011*) and a cationic leak current (*Zhang and Linden, 2006*; see also *Raman et al., 2000*). Both currents are active during the initial depolarization, and both increase in amplitude as the voltage is ramped down and the driving force increases. With further hyperpolarization, however, the L-type current deactivates, decreasing the current amplitude, until only the leak current remains. Application of TBOA increases the current amplitude between -80 and -35 mV in all four groups, consistent with both currents being potentiated by mGluR1/5 activation (*Figure 3D*). The magnitude of the current increase varied widely across cells, and differences were not statistically significant despite different mean values (repeated-measures ANOVA p=*0.66*, males n=10, females n=8). Likewise, among *Gabrb3* mutants, neither male nor female mutants were statistically different from wild-type mice (males p=*0.92*, n=11; females p=*0.92*, n=9).

Group I mGluR antagonists partly blocked the TBOA-induced current increase in each group. Wild-type males and females showed a similar response to application of mGluR1/5 antagonists (*Figure 3E*, +/+ males *vs.* females p=*0.99*). In fact, mGluR1/5 antagonists blocked little TBOA-induced current evoked below -50 mV, a voltage-dependence consistent with that of L-type Ca channels (*Zheng and Raman, 2011*). Mutant females were indistinguishable from wild-type females (p=*0.96*), but mutant males showed a trend toward more current that was sensitive to mGluR1/5 antagonists (p=*0.086 vs. +/+* males). In mutant males, the TBOA-induced current was blocked by mGluR1/5 antagonists at a wider range of voltages than the other groups, suggesting that in these mice transporter blockade also increased the leak current in an mGluR1/5-dependent manner.

Since the sex- and mutation-specific differences in mGluR1/5-dependent current are largely eliminated by blockade of glutamate uptake, then they may be independent of differences in group I mGluR expression. We tested this idea with Western blots for both mGluR1 and mGluR5 in the cerebellar nuclei of wild-type mice (both *Gabrb3* sibling +/+ and C57BL/6) and *Gabrb3* m-/p+ mice of both sexes. Indeed, neither mGluR1 nor mGluR5 protein levels differed significantly between wild-type males and females, or between wild-type males and mutant males (*Figure 3F,G*). Expression of mGluR1 did not differ between any of the four *Gabrb3* groups (n=3, one-way ANOVA p=*0.61*); the lack of a detectable sex difference was confirmed in C57BL/6 wild-type mice (n=6, p=*0.61*). Likewise, expression of mGluR5 did not differ among any of the *Gabrb3* groups, n=3, p=*0.82*) or between C57BL/6 males and females (n=6, p=*0.19*). In conjunction with the electrophysiological data in TBOA, these results support the idea that the smaller evoked mGluR1/5-dependent currents observed in wild-type males, relative to wild-type females or mutant males, arise largely from access of glutamate to group I mGluRs.

In the same cells, we examined evoked excitatory synaptic transmission and its alteration by TBOA. Afferents to CbN cells were stimulated at 100 Hz, in SR95531, strychnine, and CPP, as above. AMPARs were left unblocked so that AMPAR-mediated EPSCs could serve as evidence that glutamate release was evoked. The initial peak EPSC was decreased in TBOA by about 50% in all groups but wild-type females (+/+ males p=*0.008*, n=7; m-/p+ males p=*0.047*, n=7; +/+ females p=*0.65*, n=6; m-/p+ females p=*0.006*, n=8). The EPSC amplitude was not further changed by application of group I mGluR antagonists, however (*Figure 4A,B*, *vs.* TBOA, +/+ males p=*0.18*, m-/p+ males p=*0.97*, +/+ females p=*0.27*, m-/p+ females *p=0.66*), suggesting that glutamate accumulation induced by TBOA may decrease EPSC amplitude by acting at other receptors, e.g., group II/III mGluRs, or by directly desensitizing AMPA receptors. Additionally, attributes of these targets differ between the sexes, as well as between wild-type and mutant females.

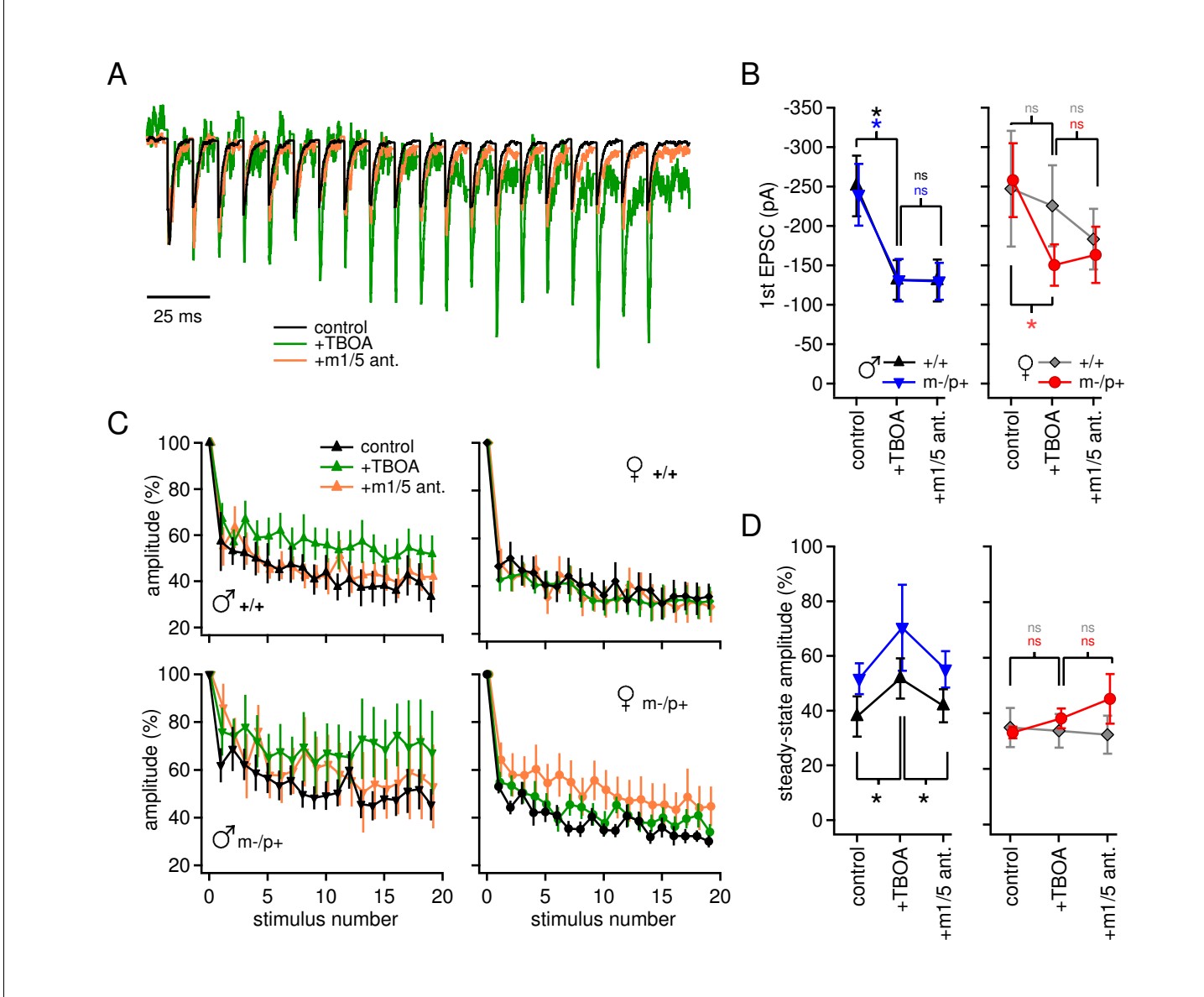

**Figure 4.** Group I mGluR-dependent and -independent effects of TBOA on evoked EPSCs. (A) Sample traces from a CbN cell from a m-/p+ male of EPSCs elicited by 100 Hz stimulus trains, in control, TBOA, and TBOA + mGluR1/5 antagonist(s), normalized to the peak of the first EPSC. (B) First EPSC amplitude in control, TBOA, and TBOA + mGluR1/5 antagonist(s). Color code indicates statistical differences, 'ns,' non-significant. (C) EPSC amplitude for EPSCs evoked at 100 Hz in control, TBOA, and TBOA + mGluR1/5 antagonist(s). (D) The steady-state amplitude, measured as the mean of the last 5 EPSCs normalized to the first EPSC, for males (left) and females (right) in control, TBOA, and TBOA + mGluR1/5 antagonist(s).

The total amplitude of train-evoked EPSCs in these experiments was composed of summed AMPAR and mGluR responses, and increases in amplitude relative to the first EPSC could result from changes in either current. When normalized to the initial amplitude, EPSCs from all four groups depressed by about 60% (*Figure 4C,D*) by the end of the train. In wild-type males, however, TBOA reduced synaptic depression (+/+ males, control 38 ± 7%, TBOA 52 ± 7%, p=*0.012*), and, unlike the change in initial EPSC amplitude, the effect of TBOA on synaptic depression was blocked by group I mGluR antagonists (+/+ males with antagonist 42 ± 6%, p=*0.002*). This observation lends support to the idea that the lack of detectable evoked mGluR1/5-dependent current in wild-type males depends on intact glutamate transport. The results in mutant males resembled those of wild-type males, in that mean synaptic depression was reduced in TBOA, but responses were more variable

across cells, occasionally even showing facilitation (m-/p+ males, control 49 ± 7%, TBOA 70 ± 16%, p=0.14). In mGluR1/5 antagonists, currents were restored to control values (m-/p+ males, with antagonist 55 ± 7% p=0.20 vs. control). In contrast, in both female groups, TBOA had no detectable effect on synaptic depression (+/+ females p=0.81, m-/p+ p=0.22). Thus, although many variables interact to generate the profiles of currents, the data are consistent with sex-specific differences in mGluR currents activated when glutamate transporters are blocked: (1) an mGluR1/5-*independent* decrease in initial evoked EPSC amplitude, evident in both male groups as well as in mutant females, and (2) an mGluR1/5-*dependent,* sex-specific decrease in synaptic depression in males only, which is relatively unaffected by the *Gabrb3* mutation.

The observation of sex differences in basal synaptic properties, as well as the unanticipated link between the loss of an allele encoding a $GABA_A R$ subunit and changes in group I mGluR-mediated responses, raises the possibility that other aspects of excitability also differ between the sexes and/or are differentially regulated by the mutation. We therefore measured spontaneous firing rates in CbN cells, with fast excitatory transmission blocked by 5 µM DNQX and 10 µM CPP. Indeed, CbN cells fired more slowly in wild-type males (65 ± 7 spikes/s, n=14) than in females (98 ± 9 spikes/s, n=16; p=0.02), again reflecting a sex difference in basic CbN cell properties (*Figure 5A,B*). CbN cells from mutant males tended to fire somewhat faster than from wild-type males (92 ± 9 spikes/s, n=11, p=0.08), whereas cells from mutant females had the same firing rates as those from wild-type females (96 ± 10 spikes/s, n=20, p=0.9). Action potential half-widths, however, were indistinguishable among all four groups (*Figure 5B*, p=0.3). Thus, the sex differences in basal cerebellar physiology and in response to the *Gabrb3* m-/p+ mutation may extend beyond synaptic activation of mGluRs. These differences in CbN cell spontaneous rates mirror the differences in synaptic responses, raising the possibility that chronic activation of group I mGluRs may contribute a depolarizing current that affects firing. The experiments of *Figure 3*, however, demonstrated that holding currents at -40 mV were not different across the four groups of mice (+/+ males -565 ± 40 pA; m-/p+ males -491 ± 59 pA; +/+ females -406 ± 83 pA; m-/p+ females -424 ± 49 pA, p=0.26). It therefore seems unlikely that the slower firing rates in wild-type males arose from mGluR1/5 activation, and instead are more likely to reflect differences in intrinsic currents.

In addition to reducing the time course of IPSCs, counteracting tonic inhibition, and altering excitatory synaptic depression, group I mGluRs may influence the output spiking of CbN cells directly. Specifically, after periods of spike suppression either by hyperpolarization or by trains of IPSCs, CbN neurons have been shown to increase their firing above their intrinsic rates for a few hundred milliseconds. This phenomenon of 'prolonged rebound firing' depends partly on recovery of intrinsic conductances (*Aman and Raman, 2007*; *Sangrey and Jaeger, 2010*; *Tadayonnejad et al., 2010*) and partly on mGluR1/5 potentiation of L-type Ca currents (*Zheng and Raman, 2011*). If the increase in train-evoked mGluR1/5-dependent current observed in neurons from male mutants and all female mice leads to a greater potentiation of L-type currents, the extent of CPCCOEt-sensitive prolonged rebound firing should be greater in neurons from these animals than from wild-type male mice.

We therefore evoked prolonged rebound firing before and after application of CPCCOEt in CbN cells from wild-type and *Gabrb3* m-/p+ mice. In CPP and DNQX, CbN cells were current-clamped and enough hyperpolarizing holding current was applied to reduce regular intrinsic firing to a frequency near 40 Hz. When 100-Hz, 500-ms stimulus trains were applied, firing either slowed or ceased, consistent with the outward currents seen in voltage clamp. In all cells, firing rates for the 300 ms just after the stimulus were faster than firing rates during the 500 ms just before the stimulus (*Figure 5C*). To quantify these changes and account for small differences in the pre-stimulus baseline firing rates across cells, we calculated the percent change in rate from the ratio of the post-stimulus to the pre-stimulus rate (*Figure 5D*). Consistent with an upregulation of mGluR1/5 responses, the firing rate acceleration in CbN cells was greater in m-/p+ than wild-type males. Moreover, in the same cells, this difference disappeared when group I mGluRs were blocked. In females, cells from both wild-type and mutant mice showed prolonged rebound firing that was reduced on average by CPCCOEt.

To quantify the contribution of group I mGluRs, the difference in the percent change in control and CPCCOEt solutions was calculated (*Figure 5E*). This mGluR1/5-dependent increase in prolonged rebound firing did not differ between female wild-type and m-/p+ mice (+/+ 39 ± 35%,

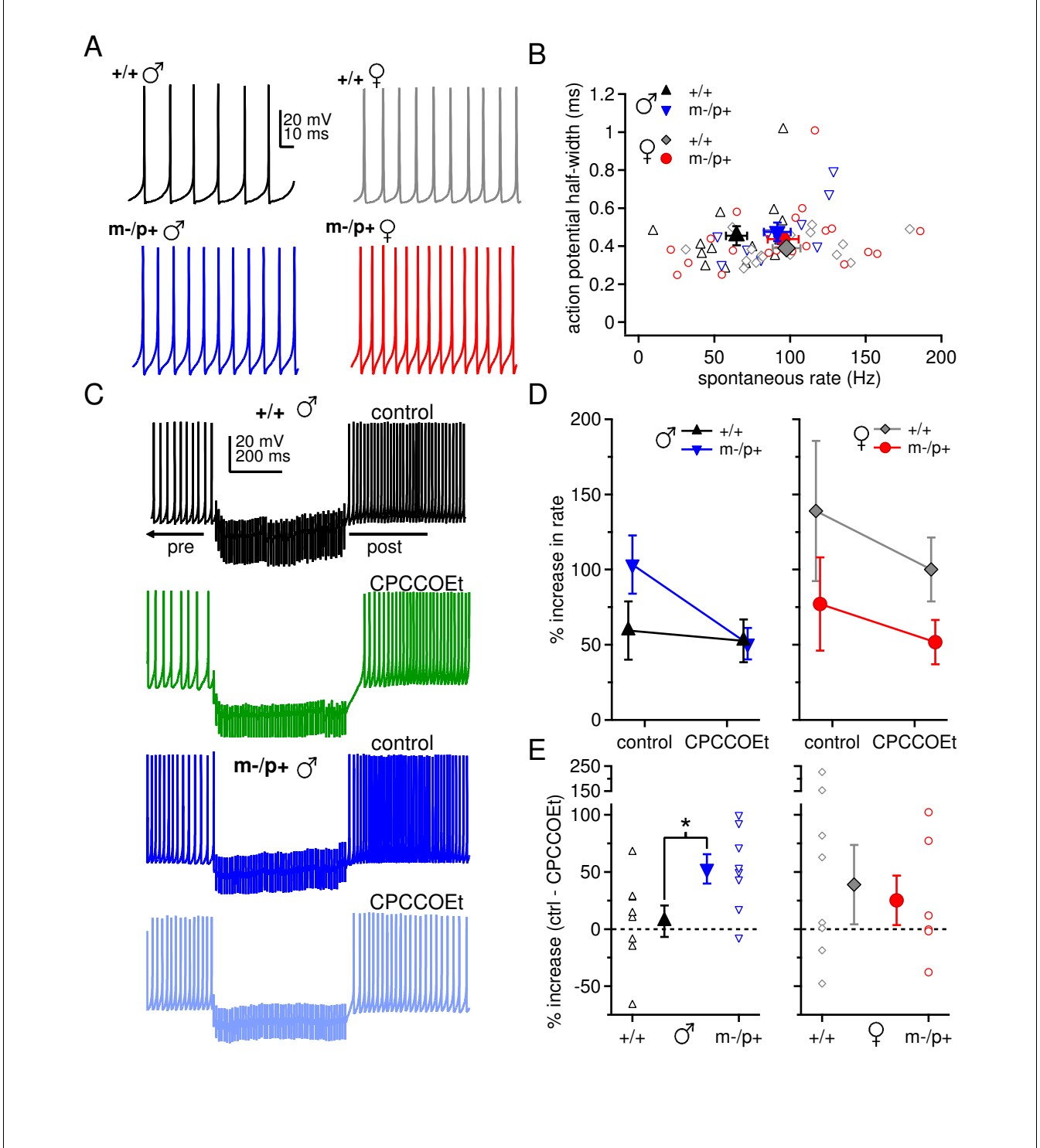

**Figure 5.** Sex differences in spontaneous firing rates and evoked rebound firing, and sex-specific responses to the *Gabrb3* m-/p+ mutation. (A) Sample traces of spontaneous action potentials in CbN cells. (B) Action potential half-width *vs.* spontaneous firing rate for all cells from all groups. Open symbols, individual cells; closed symbols, mean values. (C) Action potentials in a CbN cell, interrupted by a 500-ms, 100-Hz net inhibitory stimulus train, before and after application of CPCCOEt, for wild-type and m-/p+ CbN cells from males. To facilitate comparison, pre-stimulus firing was kept near 40 Hz (below the spontaneous rate) with constant holding current. (D) Difference between post- and pre-stimulus rates as a percentage of the pre-stimulus rate, before and after application of CPCCOEt for CbN cells from males (left) and females (right). (E) The percent increase in rate that depends on mGluR1/5, calculated from cell-by-cell differences between rate increases ± CPCCOEt in C, for males (left) and females (right). Dotted lines, 0%.

m-/p+ 25 ± 22%, p=0.74). In contrast, in CbN cells from wild-type males, only 7 ± 14% of the firing rate increase after inhibition can be attributed to group I mGluRs, while for cells from m-/p+ males, 53 ± 13% is CPCCOEt-dependent (p=0.03). This small effect of CPCCOEt on prolonged rebound firing in wild-type males correlates with the low magnitude of evoked mGluR1/5-dependent current in these animals. The larger CPCCOEt effect in mutant males is consistent with the idea that the upregulation of the mGluR1/5 response indeed generates a larger potentiation of L-type currents. In females, a CPCCOEt effect was evident in both wild-type and mutant mice, consistent with the voltage-clamp measurements, but responses were generally more variable than in males, suggesting that factors in addition to mGluR1/5 activation may affect prolonged rebound firing in females.

The sex differences both in basal synaptic physiology of cerebellar output neurons and in their responses to the *Gabrb3* mutation raise the question of whether and how these differences are manifested in cerebellar behavior. Since several genetic disruptions that alter cerebellar output have been demonstrated to change performance on the accelerating rotarod (*Caston et al., 1995*; *Lalonde et al., 1995*; *Gerlai et al., 1996*; *Levin et al., 2006*; *Galliano et al., 2013*), we tested mice from all four groups on this motor task. Experiments were done on P22 mice (on day 1 of training) to permit valid correlations of behavior with electrophysiological data. Notably, despite the differences in basal physiology, wild-type males and females performed the task equivalently (p=0.21). This observation demonstrates that the larger mGluR1/5 responses and faster firing rates seen in CbN cells of females do not directly translate to enhanced rotarod performance per se, instead illustrating that wild-type brains of different sexes may use different synaptic and cellular mechanisms to achieve a common behavioral output (*De Vries, 2004*).

Comparing the responses of mutant mice to sex-matched controls showed that mutant males performed indistinguishably from wild-type male siblings: both groups remained on the rotarod for a similar duration on Day 1 (+/+ 102 ± 14; m-/p+ 101 ± 10 s, p=0.9, for all groups n=8) and improved their performance by Day 7 (+/+ 142 ± 11; m-/p+ 133 ± 12 s; change over training assessed by repeated measures ANOVA, +/+, p=0.02; m-/p+, p=0.03). In contrast, female m-/p+ mice tended to remain on the rod for a *longer* time on Day 1 than wild-type siblings (+/+ 78 ± 6; m-/p+ 135 ± 11 s, p=0.001), and their performance remained relatively constant over 7 days, unlike wild-type females, whose latency to fall increased (Day 7 latency, +/+ 130 ± 13; m-/p+ 116 ± 13 s; repeated measures ANOVA, +/+, p<0.00; m-/p+, p=0.3, *Figure 6A*). Given the long but replicable latency to fall on Day 1 of female mutants, the lack of increase in fall latency over training indicates an enhanced initial ability coupled with either a failure to improve, or a saturation of performance that precludes further improvement. A similarly enhanced performance by naïve mice has been reported previously for other mutations (*Vitali and Clarke, 2004*).

To assess differences between mutant and wild-type mice of the same sex, we quantified the change of each group over the course of training (Δ latency) by subtracting the initial latency to fall on Day 1 from that of the final 'steady-state' latency, given as the mean latency on Days 5–7, and compared this value between the relevant groups. Wild-type and mutant males performed similarly, with Δ latencies of 46 ± 17 s and 43 ± 12 s, respectively (p=0.9). In females, however, wild-type mice increased their latency by 71 ± 13 s, while the latency of mutant mice decreased by 8 ± 11 s (*Figure 6B*, p<0.001). The differences observed in rotarod performance was independent of body mass, as mutant females weighed the same as wild-type females (*Figure 6—figure supplement 1*, p=0.23). In weanling mice, therefore, the *Gabrb3* m-/p+ mutation had no discernible effect on rotarod performance in males, but altered it in females. Previous work (*DeLorey et al., 2011*) has shown the opposite results in adult mice (10–35 weeks old): m-/p+ males, but not females, fail to increase their latency to fall. We therefore tested older mice, which confirmed that the effect of the mutation varies with age as well as sex, with mutant male performance degrading in adult animals, and adult mutant female performance resembling that of wild-type (*Figure 6—figure supplement 2*).

The present behavioral data therefore suggest that the mutation-dependent electrophysiological changes in weanling males were likely to be part of an effective compensation, preventing them from displaying behavioral abnormalities. Interestingly, however, the same mutation, which led to fewer, more variable changes in the measured electrophysiological parameters, altered rotarod behavior in females, suggesting that changes in other cerebellar signals or effects elsewhere in the brain ultimately produced a deviation from normal. Nevertheless, with a global mutation, the

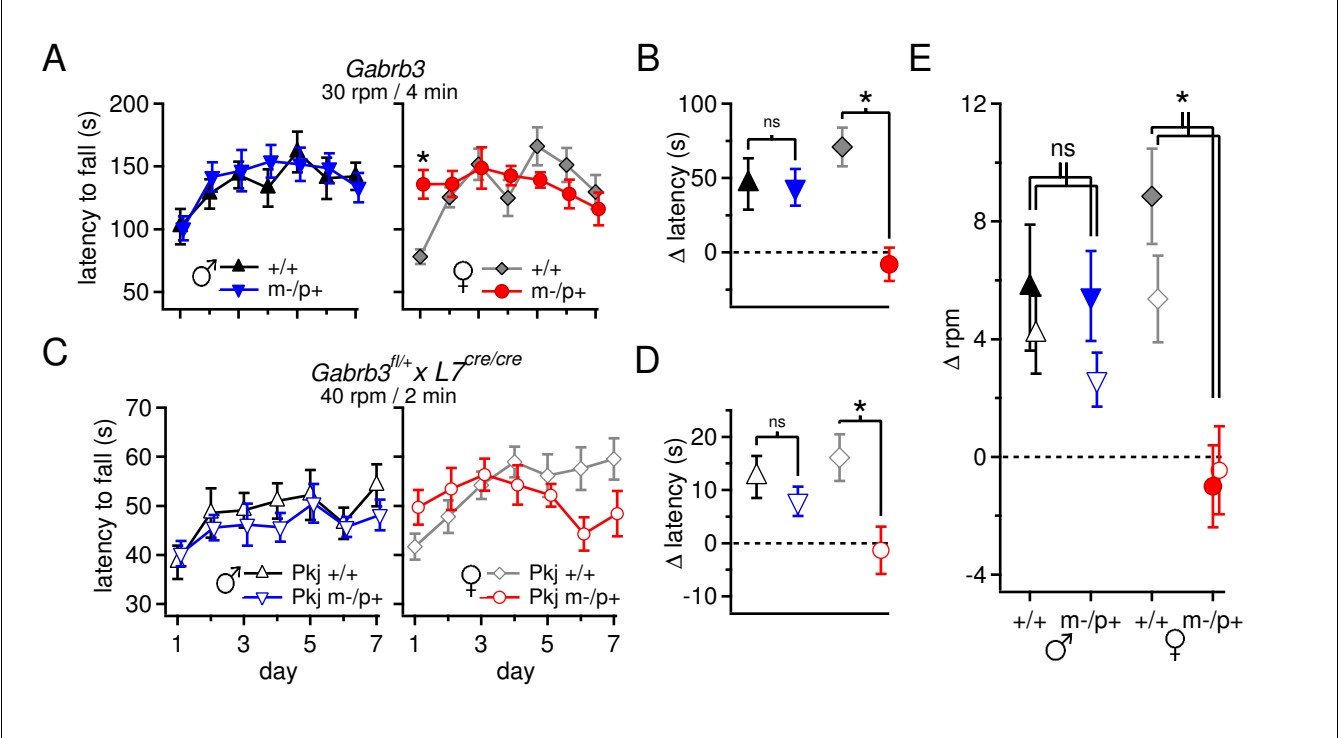

**Figure 6.** Performance on the accelerating rotarod varies with sex and *Gabrb3* and depends on GABA$_A$R β3 expression only in Purkinje cells. (A) Latency to fall *vs.* training day for *Gabrb3* P22 males (left) and females (right). (B) Change in latency, calculated as the difference between the Day-1 fall latency and mean Day 5–7 fall for all four groups. Symbol color code as in (A). (C) Latency to fall *vs.* training day for P22 Purkinje-specific *Gabrb3* P22 males (left) and females (right). (D) Change in latency, calculated as the difference between the Day-1 fall latency and mean Day 5–7 fall for Purkinje-specific *Gabrb3* mice. Symbol color code as in (C). (E) Change in the rotation rate (in rpm) at which the mouse fell on Day-1 *vs.* mean Day 5–7 for global *Gabrb3* mice (closed symbols) and Purkinje-specific *Gabrb3* mice (open symbols).

The following figure supplements are available for figure 6:

**Figure supplement 1.** Mouse weight does not account for differences in motor learning.

**Figure supplement 2.** Differences in performance on rotarod change over development.

**Figure supplement 3.** Selection of rotarod protocols that permit an increase in fall latency over training for transgenic mice on a 129S background.

observed changes in behavior cannot be directly attributed to changes in synaptic input to the cerebellar nuclei. Therefore, based on the observation that GABA$_A$R β3 is expressed more strongly in the cortex than the nuclei (*Figure 1G*), we repeated the tests of rotarod performance in mice lacking maternal *Gabrb3* in Purkinje cells only (referred to as 'Purkinje-specific mutant' or 'Pkj m-/p+', and wild-type siblings as 'Pkj +/+'). These mice, however, were of a different strain than the global mutants; the former were bred on a 129S background whereas the latter were on a C57BL/6 background. Subjecting Pkj +/+ mice to the acceleration protocol used previously (30 rpm over 4 min) did not result in a prolonged latency to fall (*Figure 6—figure supplement 3A*), consistent with reported observations that 129S mice are generally less capable of motor learning than C57BL/6 (*Kelly et al., 1998*; *Homanics et al., 1999*; *Contet et al., 2001*; *Võikar et al., 2001*). We therefore tested a series of acceleration protocols to identify one in which wild-type mice on a 129S background improved their performance over the duration of training (*Figure 6—figure supplement 3B*).

Like the global mutants, Purkinje-specific mutant males did not differ from wild-type males (Δ latency for Pkj +/+ males 12 ± 4 s, n=10; Pkj m-/p+ males 8 ± 3 s, n=11; p=0.40), while Purkinje-specific mutant females had a significantly reduced Δ latency when compared to wild-type females

(Δ latency Pkj +/+ females 16 ± 4 s, n=10; Pkj m-/p+ females -1.4 ± 4 s, n=9; p=*0.004*, *Figure 6C, D*). To directly compare the performance of each group in the Purkinje-specific *Gabrb3* experiment with the corresponding group in the global *Gabrb3* experiment, despite the different acceleration speeds of the two protocols, we plotted the change in performance (days 5–7 minus day 1) as the change in revolutions per minute (Δ rpm) at the time of fall (*Figure 6E*). The results indicate that the Δ rpm across all four groups (wild-type and mutant males and females) for Purkinje-specific *Gabrb3* mice strongly resembles that of the global *Gabrb3* mutation. Thus, disrupting *Gabrb3* expression in Purkinje cells alone is sufficient to cause the behavioral alterations observed in mutant females, but not males.

The question remains, however, whether loss of the maternal allele only in Purkinje cells is sufficient to elicit the electrophysiological changes observed in the global mutants. We therefore, repeated the experiment of *Figure 1A–D* in the Purkinje-specific *Gabrb3* mice. As before, a basal sex difference was evident, as the tonic outward current produced by a 100-Hz train stimuli was higher in wild-type males than in wild-type females (*Figure 7A-7D*, tonic current for last IPSC, Pkj +/+ males 7.3 ± 2.5%, n=8; Pkj +/+ females 1.5 ± 2.1%, n=9; repeated measures ANOVA p=*0.06*). In addition, the Purkinje-specific mutant males had less tonic current than wild-type males (Pkj m-/p+ males 0.04 ± 1.5%, n=8, p=*0.02 vs.* Pkj +/+), consistent with an upregulation of mGluR1/5-dependent current. In contrast, wild-type and mutant females had similar levels of tonic current (Pkj m-/p+ females -2.8 ± 2.0%, n=7, p=*0.14 vs.* Pkj +/+). In addition, measurements of IPSC decay kinetics were similar in the Purkinje-specific and global mutants. IPSCs in CbN cells on the 129S background were again slower in wild-type males than in wild-type females (*Figure 7E*, Pkj +/+ males $\tau_{decay}$ = 3.1 ± 0.6 ms, n=6; Pkj +/+ females 1.8 ± 0.2 ms, n=7, p=*0.02*), and neither mutant males nor mutant females differed from their sex-matched controls (Pkj m-/p+ males 2.2 ± 0.6 ms, n=3, p=*0.19 vs.* Pkj +/+; Pkj m-/p+ females 1.5 ± 0.2 ms, n=5, p=*0.61 vs.* Pkj +/+). Together, the data illustrate that the Purkinje-specific mutation reproduces the enhanced tonic inward current in mutant males but not females, as well as the cerebellar behavioral alteration in mutant females but not males. These results support the hypothesis that the upregulation of evoked mGluR1/5-dependent current in CbN cells of mutant males arises indirectly

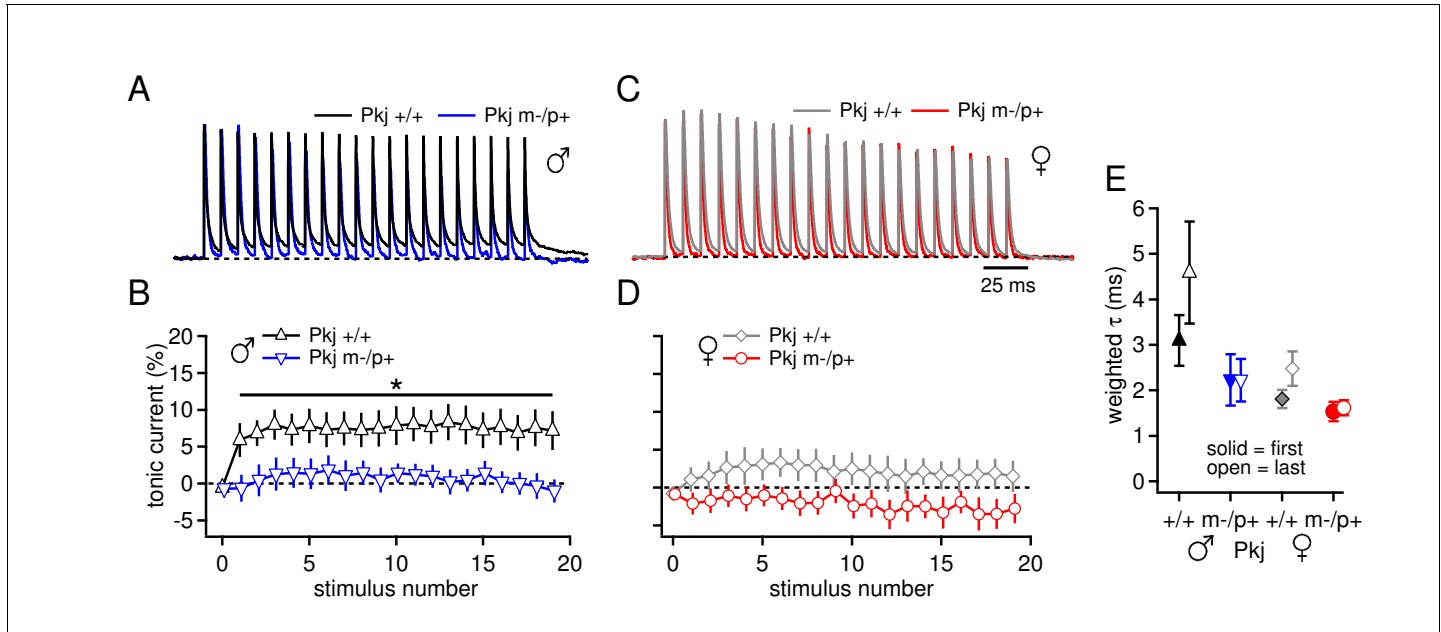

**Figure 7.** Differences in CbN cell tonic current depend on GABA_AR β3 expression only in Purkinje cells. (**A**) 100-Hz trains of synaptic currents evoked as in *Figure 1* in CbN cells from Purkinje-specific *Gabrb3* +/+ and m-/p+ male mice, normalized to the first peak. Dotted line, baseline holding current. (**B**) Mean amplitudes of tonic synaptic currents as a percentage of the first peak evoked current *vs.* stimulus number for cells from Pkj +/+ and Pkj m-/p+ male mice. Dotted line, 0% current. (**C, D**) As in A, B, but for cells from +/+ and m-/p+ female mice. (**E**) Solid symbols: weighted $\tau_{decay}$ for IPSCs from a single stimulus for each group of CbN cells from Purkinje-specific *Gabrb3* mice. Open symbols: weighted $\tau_{decay}$ for the last IPSC in the train.

from the reduced GABA$_A$R β3 subunit in the cerebellar cortex, and strongly suggest that this upregulation serves a compensatory role in cerebellar motor learning tasks.

## Discussion

These data provide evidence for a sex difference in basal cerebellar synaptic physiology in weanling mice, including differences in synaptic excitation at the level of group I metabotropic glutamate receptors, synaptic inhibition at the level of IPSC kinetics, and intrinsic properties at the level of spontaneous firing rates. The response to an autism-linked mutation likewise differs between the sexes. With the *Gabrb3* m-/p+ mutation, which is predicted to increase inhibitory drive from Purkinje cells, CbN neurons in male mice upregulate their mGluR1/5 responses to synaptic stimulation, whereas females do not. This change in the mGluR1/5 response appears to be a direct response to changes in Purkinje cell input, since restricting the m-/p+ mutation to Purkinje cells is sufficient to elicit it, and it also appears compensatory, since mutant male mice perform indistinguishably from wild-type males in a motor learning task that involves the cerebellum. In contrast, mutant females, which do not upregulate their mGluR1/5 responses, perform differently from wild-type females. Thus, sex differences in electrophysiological properties are present in a brain region in which such differences have not been extensively investigated. The sex-specific synaptic physiology in turn provides distinct backgrounds on which responses to genetic alterations occur.

### Deficits and compensation induced by the *Gabrb3* m-/p+ mutation

The reduced expression of GABA$_A$R β3 leads to an upregulation of evoked mGluR1/5-dependent inward currents in large premotor cells of the CbN in male but not female mutant mice. Such indirect effects of reduced GABA$_A$R β3 expression could be an exacerbation of or a compensation for disrupted synaptic physiology elsewhere in the circuit. Since the GABA$_A$R β3 subunit is expected to prolong IPSCs (*Hinkle and Macdonald, 2003*) in cells that express it, i.e., Purkinje and cerebellar granule cells (*Figure 1F*; *Laurie et al., 1992*; *Fritschy and Mohler, 1995*; *Hörtnagl et al., 2013*), reduced GABA$_A$R β3 expression in m-/p+ mice is predicted to disinhibit granule and Purkinje cells during behaviors that normally engage cerebellar cortical interneurons. The simplest mode of counterbalancing the resulting elevation of inhibition to CbN cells would be by increasing the net excitatory drive and/or raising intrinsic excitability. Indeed, evidence for both changes were observed in male m-/p+ mice. The larger mGluR1/5-dependent currents evoked by stimulation of excitatory afferents serve to counteract much of the tonic outward current elicited by concurrent stimulation of inhibitory afferents, and the increased activation of group I mGluRs is sufficient to raise excitability, as evident by the facilitation of prolonged rebound firing relative to wild-type cells. In addition, the increase in spontaneous firing rates in mutant males, which occurs without a measurable increase in holding current when cells are voltage-clamped, suggests that one or more intrinsic ion channels may also be modulated by the mutation. Independently of mechanism, the observed increase in mGluR1/5-dependent current and intrinsic excitability in male *Gabrb3* m-/p+ CbN cells is well suited to compensate for the predicted increase in inhibitory drive (*Figure 8*, top). Indeed, the results from the rotarod experiments suggest that this compensation successfully prevents mutant males from displaying altered motor behavior.

Males and females respond differently to the m-/p+ mutation; although GABA$_A$R β3 protein levels are reduced in m-/p+ females, CbN cells from wild-type females already have elevated spontaneous firing rates and relatively large evoked mGluR1/5 responses, and no further changes in these parameters are seen with the mutation (*Figure 8*, bottom). The difference in rotarod behavior between wild-type and mutant weanling females in the global and Purkinje-specific *Gabrb3* mice suggests that this absence of an indirect response has behavioral consequences.

The disruption in rotarod performance in mutant females is unusual: naïve mutant females show prolonged latency to fall on Day 1, and fall latency is not prolonged by subsequent training. Interestingly, previous studies have reported enhanced initial performance on accelerated rotarod by naïve mice with a mutation in a protein repair methyltransferase; these mice were also hyperactive (*Vitali and Clarke, 2004*). Hyperactivity has been observed in homozygous mutant *Gabrb3* mice (*DeLorey et al., 1998*; *Liljelund et al., 2005*), but not in m-/p+ (*Liljelund et al., 2005*) or mixed paternal/maternal heterozygotes (*DeLorey et al., 1998*). Neither of these studies separated mice by sex, however, leaving open the possibility that m-/p+ females are more hyperactive than m-/p+

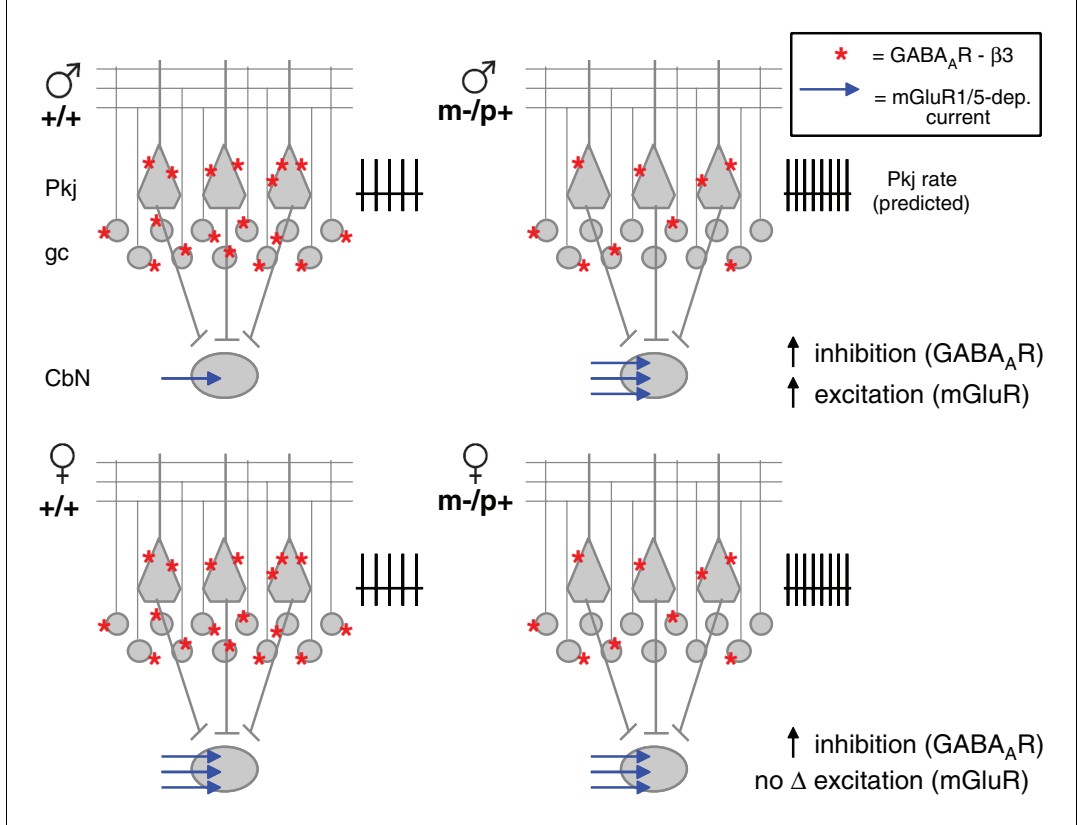

**Figure 8.** Diagram of sex differences in mGluR1/5 and compensatory changes in males with loss of GABA_AR β3. (**Top:**) Expected relative levels of GABA_AR β3 expression (red asterisks) in the cerebellar cortex, and mGluR1/5-dependent current amplitude (blue arrows) in large CbN cells. For simplicity, only Purkinje cells (Pkj), granule cells (gc), and large CbN cells are depicted. The m-/p+ mutation reduces β3 expression, predicting a disinhibition of granule and Purkinje cells. The resulting increase in Purkinje cell firing rates should increase inhibition of CbN cells. (**Top:** ) Mutant males counteract the predicted increase in inhibition with increased via mGluR1/5-dependent inward currents. (**Bottom:** ) Wild-type females have more mGluR1/5-dependent current than wild-type males and presumably balance excitation and inhibition through other means. Mutant females do not up-regulate mGluR1/5-dependent current, and apparently do not compensate for the increased inhibition.

males, which may contribute to their prolonged fall latency early in the task without improvement over the training period.

The change in evoked mGluR1/5 responses upon loss of GABA_AR β3 illustrates the extensive plasticity of the cerebellar circuit, which often appears homeostatic in response to physiological perturbations and can be at least partly compensatory in response to genetic disruptions (*Zheng and Raman, 2010*). For example, in the *pcd* and *lurcher* mutations, loss of inhibitory input to CbN neurons from degeneration of Purkinje cells triggers structural and physiological changes that increase the efficacy of the residual inhibition (*Garin et al., 2002*; *Sultan et al., 2002*; *Linneman et al., 2004*). Similarly, in Purkinje neurons, deletion of *Scn8a*, which encodes the voltage-gated Na channel α subunit Na_V1.6, slows Purkinje cell firing but also increases facilitation of afferent parallel fibers, elevating excitatory drive to Purkinje cells (*Levin et al., 2006*). Finally, eliminating large conductance Ca-activated K channels, expected to reduce Purkinje cell firing, results in a reduction of climbing fiber activity, which should potentiate excitatory inputs to Purkinje cells (*Chen et al., 2010*).

## Sex differences in the cerebellum

Sex differences are evident in three dimensions of CbN cell physiology, including synaptic excitation, synaptic inhibition, and intrinsic properties. Specifically, males and females have (1) different magnitudes of evoked mGluR1/5-dependent currents, possibly owing to differential access of glutamate

to receptors, (2) different IPSC decay kinetics, and (3) different spontaneous firing rates. Although the etiology of these differences has yet to be fully examined, they seem likely to arise from a sex difference in neuronal properties rather than from modulation by circulating gonadal hormones, since they are present in pre-pubertal mice. This idea is compatible with other reports of sexually dimorphic molecular and cell biological attributes of cerebellar neurons in juvenile rodents (*Dean and McCarthy, 2008*). For instance, calbindin expression in Purkinje neurons is higher in weanling females than males; this difference is determined by sex chromosome complement, regardless of gonadal sex (*Abel et al., 2011*). Another example of pre-pubertal cerebellar sex differences comes from rat pups, in which the inflammation-associated prostaglandin PGE2 can activate aromatase, and the resulting estradiol synthesis within the cerebellum leads to reduced Purkinje cell dendritic arborization (*Sakamoto et al., 2001*). A decrease in Purkinje cell capacitance, as a proxy for arborization, can be mimicked by exogenous estradiol in males but not females, even in juvenile rodent brains (*Dean et al., 2012*). Precedent also exists for the idea that genetic mutations in the cerebellum are differentially manifested between the sexes; the *reeler* and *staggerer* mutations, which both lead to Purkinje cell degeneration, have earlier and/or more extreme effects in heterozygous males than females, with differences evident in animals as young as one month old (*Hadj-Sahraoui et al., 1996*; *Doulazmi et al., 1999*).

## Group I mGluRs in the cerebellar nuclei

Metabotropic glutamate receptors and their downstream targets have varied effects in the cerebellar nuclei, such that changes in mGluR1/5-dependent current may alter cerebellar output. Potentiation of L-type Ca channels by group I mGluRs can accelerate prolonged rebound firing (*Zheng and Raman, 2011*), and group I mGluRs can also generate a slow EPSC (*Zhang and Linden, 2006*). In the present study, the relative amplitudes of inward and outward synaptic currents reported are specific to the experimental conditions. In vivo, processed sensory signals from mossy fibers are expected to excite CbN cells and activate the granule cell-Purkinje cell pathway that concurrently releases GABA onto the same CbN neurons. Thus, the amplitude of mGluR1/5-dependent currents relative to IPSCs is likely to regulate CbN cell firing in vivo. Moreover, the role of group I mGluRs in the cerebellum may differ between the sexes, as has been shown in other brain regions (*Boulware et al., 2005*; *Tabatadze et al., 2015*).

Group I mGluRs are also implicated in long term synaptic plasticity in the cerebellar nuclei. Specifically, induction of long-term depression (LTD) of mossy fiber EPSCs in CbN cells requires activation of mGluR1 (*Zhang and Linden, 2006*). Conversely, long-term plasticity (LTP) of mossy fiber EPSCs in the cerebellar nuclei is inhibited by flux through L-type Ca channels (*Pugh and Raman, 2006*; *2008*; *Person and Raman, 2010*), which are regulated by group I mGluRs (*Zheng and Raman, 2011*). In all these studies of cerebellar physiology in juvenile mice, however, data were gathered from both sexes and analyzed together. Considered with the present work, previous findings raise the possibility that LTD and LTP may be differentially induced in CbN cells of *Gabrb3* m-/p+ males relative to wild-type males, or even in male *vs.* female wild-type mice.

## Neurodevelopmental disorders and mGluRs

Accumulating evidence suggests a role of mGluRs in several neurodevelopmental disorders. For instance, mGluR5 has been implicated in multiple models of intellectual disability and autism, including fragile X syndrome, tuberous sclerosis complex (TSC), and Phelan McDermid syndrome (*D'Antoni et al., 2014*). Male mice deficient in TSC show deficits in mGluR-mediated CA1-LTD (*Ehninger et al., 2009*; *Auerbach et al., 2011*). *Tsc1* and *Tsc2* mice (male and female, pooled) both have impaired hippocampal learning and memory (*Goorden et al., 2007*; *Ehninger et al., 2008*); interestingly, loss of *Tsc1* from Purkinje cells alone in male mice mimics many phenotypes of the disorder (*Tsai et al., 2012*). Additionally, male mice deficient in the (X-linked) fragile X mental retardation gene (*Fmr1*), whose protein product FMRP binds mRNA and regulates translation (*Feng et al., 1997*; *Laggerbauer et al., 2001*; *Li et al., 2001*), display exaggerated mGluR-mediated hippocampal CA1-LTD (*Huber et al., 2002*). Both global and Purkinje cell-specific *Fmr1* knockout mice show enhanced parallel fiber LTD in Purkinje cells, as well as deficits in delay eyelid conditioning, a cerebellar behavior (*Koekkoek et al., 2005*). The present data from *Gabrb3* m-/p+ mice provide further support for the general principle of modulation of mGluRs affecting phenotypes associated with

ASD and intellectual disability, while raising the possibility of differential effects in males and females.

## Materials and methods

### Subjects and genotyping

All procedures conformed to institutional guidelines and were approved by the Institutional Animal Care and Use Committee of Northwestern University. Mice were housed with a 14:10 light:dark cycle with access to food and water *ad libitum*. Female B6;129-*Gabrb3*$^{tm1Geh}$/J ('*Gabrb3* mice', Jackson Laboratories, Bar Harbor, ME) heterozygotes were crossed with C57BL/6J (Jackson Laboratories, Bar Harbor, ME) males to create +/+ and m-/p+ offspring. Due to limited availability of *Gabrb3* mice, male and female C57BL/6J mice were used where indicated for control experiments or to expand data sets on wild-type male and female mice.

Purkinje-specific *Gabrb3* mice were created by crossing B6;129-*Gabrb3*$^{tm2.1Geh}$/J (*Gabrb3*$^{fl/+}$) females (a kind gift of Dr. Theo Palmer, Stanford University) with B6.129-Tg(Pcp2-cre)2Mpin/J (L7$^{cre/cre}$) males (Jackson Laboratories, Bar Harbor, ME), since L7 (pcp2) is strongly expressed almost exclusively in Purkinje neurons (*Oberdick et al., 1998*; *Barski et al., 2000*). Purkinje-specific *Gabrb3* mice were genotyped by Transnetyx, Inc. (Cordova, TN). *Gabrb3* mice were genotyped either by Transnetyx, Inc or in house with primers recommended by Jackson Laboratories (GCA TCG ACA TGG TTT CTG AAG TC, GGG CTA CTG ATC TCC TCT TTC CAC, and CAG AAA GCG AAG GAA CAA AGC TG, from Integrated DNA Technologies, Coralville, IA).

### Preparation of cerebellar slices

P17-24 mice were anesthetized with isoflurane and transcardially perfused with artificial cerebrospinal fluid (ACSF) at 35–37°C containing (mM) 123.8 NaCl, 0.35 KCl, 2.6 NaH$_2$CO$_3$, 0.125 NaH$_2$PO$_4$, 1.0 glucose, 1.5 CaCl$_2$, 1.0 MgCl$_2$, and bubbled with 95 O$_2$/5% CO$_2$. Mice were decapitated, and cerebella were transferred into 35–37°C oxygenated ACSF. Parasagittal cerebellar slices (300 μm) were cut on a vibratome (Leica VT1200, Leica Microsystems Inc., Buffalo Grove, IL), incubated for 30 min at 35–37°C in oxygenated (95 O$_2$/5% CO$_2$) ACSF, and then maintained at room temperature.

### Electrophysiological recording

Recordings were made from large cells in the interpositus and the medial portion of the lateral nucleus at 35–37.5°C (*Zheng and Raman, 2009*). Patch pipettes (3–6 MΩ) were pulled from borosilicate glass on a Sutter Instruments (Novato, CA) P97 puller and filled with internal solution containing (mM) 132 K-gluconate, 5.5 Na-gluconate, 3.3 NaCl, 2.2 MgCl$_2$, 10 sucrose, 11 HEPES, 1.1 EGTA, 14 Tris creatine phosphate, 4 MgATP, and 0.3 TrisGTP, buffered to pH 7.3 with KOH. Voltage- and current-clamp recordings were made with a Multiclamp 700B amplifier and pClamp acquisition software. During recordings, slices were perfused with oxygenated ACSF solution with the following compounds, as noted: 5 μM DNQX (dinitrofiquinoxaline-2,3-dione) to block AMPA receptors, 10 μM CPP [(RS)-3-(2-carboxy-piperazin-4-yl)-propyl-1-phosphonic acid] to block NMDA receptors, 10 μM SR-95531 to block GABA$_A$ receptors, 10 μM strychnine to block glycine receptors, 1 μM tetrodotoxin (TTX) to block voltage-gated sodium channels, 50 μM DL-TBOA (DL-*threo*-β-benzyloxyaspartic acid) to block excitatory amino acid transporters, 100 μM CPCCOEt (7-(hydroxyimino)cyclopropa[*b*]-chromen-1a-carboxylate ethyl ester) to block group I mGluRs, 0.2 μM JNJ16259685 ((3,4-dihydro-2*H*-pyrano[2,3-*b*]quinolin-7-yl)-(*cis*-4-methoxycyclohexyl)-methanone) to block mGluR1, and 40 μM MPEP hydrochloride (2-methyl-6-(phenylethynyl)pyridine hydrochloride) to block mGluR5. Drugs were from Tocris Cookson (Bristol, UK) except strychnine (Sigma-Aldrich, St. Louis, MO) and TTX (Alomone Labs, Jerusalem, Israel). Other chemicals were from Sigma-Aldrich (St. Louis, MO). In experiments where either CPCCOEt or a combination of JNJ16259685 and MPEP were used to block group I mGluRs, JNJ16259685 and MPEP reversed the TBOA-induced current slightly more quickly, so ~75% of cells were treated with JNJ16259685 and MPEP, and 25% with CPCCOEt.

Synaptic currents were evoked by 250 μs current pulses at 0.01–10 mA delivered to the white matter surrounding the cerebellar nuclei, with either a concentric bipolar electrode (FHC) or a theta glass pipette filled with HEPES-buffered saline. For all voltage clamp recordings, cells were held at −40 mV, which is near the resting potential of CbN cells silenced by in TTX (*Raman et al., 2000*).

For current clamp recordings, where indicated, steady hyperpolarizing current was applied to maintain baseline firing at a desired rate. Voltages are not corrected for a measured junction potential of 10 mV. For pharmacological studies, recordings were made 5–10 min after perfusion of a drug was initiated to ensure complete equilibration. For experiments in *Figures 3* and *4*, cells were switched to current clamp during drug perfusions and allowed to fire action potentials, which prevented both the treatment-independent increase of leak current and the rundown of L-type Ca current during the experiment. For four months during the course of the experiments, construction near the animal facility produced extreme vibrations (>5000 μinches/s) that correlated with changes in wild-type synaptic physiology in wild-type males (*Figure 1—figure supplement 2*), much as the m-/p+ mutation did, indicating that cerebellar synaptic properties can be sensitive to environmental stimuli. Data obtained from mice during the period of construction were not included other analyses. The colony was re-established from newly ordered breeder pairs after vibration-inducing construction was completed.

## Western blots

Mice were perfused with ACSF, and cerebellar slices were cut as above in ice-cold (0–4°C) ACSF. Cerebellar nuclei were punched out of slices with a 200 μl pipette tip cut to ~1 mm diameter, and cerebellar cortex was separated from the brainstem. Cerebellar nuclei (7 slices per mouse) and cortex (11 slices per mouse) were separated and placed in ice cold HEPES homogenizing buffer containing (mM): 5 HEPES-KOH, pH 7.2, 320 sucrose, 5 EDTA, 1 Na orthovanadate, 50 NaF, 10 Na pyrophosphate, 20 Na glycerophosphate, 0.1 phenylmethyl-sulfonyl fluoride, plus protease inhibitor cocktail (Roche Diagnostics, Burgess Hill, UK). Each sample, for nuclei and cortex, contained tissue from two mice of the same age, sex, and genotype. Tissue homogenates were centrifuged at 1000 × g for 10 min to remove unbroken cells and nuclei. Membrane fractions were prepared by ultracentrifugation of postnuclear supernatant at 100,000 × g for 1 hr at 4°C with a fixed angle rotor (TLA100.2, Beckman Coulter, Brea, CA). Protein concentration was determined with the Bradford protein assay. Equal amounts of protein were separated on 7–10% SDS-PAGE gels and Western blotting was carried out as in *Tabatadze et al. (2013)*. Briefly, membranes were blocked with 5% nonfat milk and then incubated with one of the following primary antibodies overnight at 4°C: mouse monoclonal anti-mGluR1 (1:1000, BD Biosciences, San Jose, CA), rabbit-polyclonal anti-mGluR5 (1:1000, EMD Millipore, Darmstadt, Germany), rabbit-polyclonal anti-GABA$_A$R β3 (1:1000, Novus Biologicals, Littleton, CO) and goat polyclonal anti-β-actin (1:2000, Santa Cruz Biotechnology, Dallas, TX). Membranes were washed in TBS and then in 0.1% Tween 20 in TBS and incubated at room temperature for 1 hr with horseradish peroxidase-conjugated anti-mouse, anti-rabbit or anti-goat IgG secondary antibodies (1:1000, Vector Laboratories, Burlingame, CA). Immunoreactivity was visualized using an enhanced chemiluminescence kit (ECL Plus, Thermo scientific, Rockford, IL) and analyzed with Image J (NIH, Bethesda, MD). After visualization of mGluRs and GABA$_A$R β3, blots were stripped and probed for β-actin as a loading control. For each group, protein levels were expressed relative to β-actin in the same sample. To compare results across blots, the protein ratio for each band was normalized to the sum of the protein ratios for all samples on the blot, divided by the total number of lanes (*Degasperi et al., 2014*). Three independent experiments were run with tissue from *Gabrb3* mice (biological replicates), each with one +/+ male, one m-/p+ male, one +/+ female, and one m-/p+ female sample, and each sample was run in duplicate. Three independent experiments were run with tissue from C57BL/6 mice, each with two male and two female samples (biological replicates) to total 6 samples per sex, and each sample was run in duplicate. For comparison of cortex vs. nuclei in C57BL/6 mice, two independent samples were run for each sex and region, with no duplicates.

## Accelerating rotarod

Five cohorts of mice underwent rotarod testing (SDI rotor-rod, San Diego Instruments, San Diego, CA), consisting of four age groups: postnatal day 22 days old ('P22' groups, for both *Gabrb3* and Purkinje-specific *Gabrb3* mice), P43-47 (*Gabrb3* 'P45' group), P60-68 (*Gabrb3* 'P65' group), and P147-160 (*Gabrb3* 'P155' group). Mice were placed on the rod and allowed to acclimate for at least 30 s before the rod was accelerated at varying speeds as noted. Latency to fall was measured automatically, but was also monitored by the experimentalist. If mice grasped the

rod and rotated completely, latency to fall was counted when the mouse completed its rotation. After one trial, mice were allowed to remain in the chamber for at least 5 min, followed by another trial. The two trials were averaged for each day, and this process was repeated for a total of seven days.

## Data analysis

Data were analyzed and plotted with IGOR-Pro (Wavemetrics, Lake Oswego, OR) and are presented as mean ± SEM. Stimulus artifacts have been digitally reduced or removed in all figures. IPSCs were fit with the sum of two exponentials. Weighted time constants were calculated by fitting the sum of two exponentials to the decay phase of current, and finding the average of the time constants $\tau_{fast}$ and $\tau_{slow}$, each scaled by the fractional contribution to the total amplitude $F_{fast}$ and $F_{slow}$. Tonic currents were predicted from the measured double exponential fits to IPSCs at the beginning and end of 100-Hz trains with the following equations, which account for the gradual change in decay time over the train:

$$T_n = P_n\left(F_{fn}e^{-(8.4/\tau_{fn})} + F_{sn}e^{-(8.4/\tau_{sn})}\right) + T_{n-1}\left(F_{fn}e^{-(8.4/\tau_{fn})} + F_{sn}e^{-(8.4/\tau_{sn})}\right), \text{ with}$$

$$\tau_{fn} = \left(1 - \frac{n-1}{20}\right)\cdot\tau_{f1} + \frac{n-1}{20}\cdot\tau_{f20}, \text{ and } F_{fn} = \left(1 - \frac{n-1}{20}\right)\cdot F_{f1} + \frac{n-1}{20}\cdot F_{f20},$$

$$\tau_{sn} = \left(1 - \frac{n-1}{20}\right)\cdot\tau_{f1} + \frac{n-1}{20}\cdot\tau_{s20}, \text{ and } F_{sn} = \left(1 - \frac{n-1}{20}\right)\cdot F_{s1} + \frac{n-1}{20}\cdot F_{s20},$$

where, for the nth stimulus, $T_n$ is the tonic current, $P_n$ is the measured peak phasic current, $\tau_{fn}$ and $\tau_{sn}$ are the fast and slow decay time constants in ms, $F_{fn}$ and $F_{sn}$ are the fractional contributions of the fast and slow components of decay (which sum to 1), and 8.4 ms denotes the time after the stimulus when the tonic current is calculated for a 100 Hz train, i.e., 10 ms, less the stimulus artifact time and IPSC rise time.

Data were analyzed for statistical significance using SPSS software (IBM Corp., Armonk, NY) with a value of $p < 0.05$ considered statistically significant and indicated by an asterisk where appropriate. For all ANOVAs, significant differences were only reported if there was a significant main effect. Repeated-measures ANOVAs with Tukey post-hoc comparisons were used to assess significant differences between the wild-type sexes, and between mutants and their sex-matched controls for phasic and tonic currents, changes in holding currents upon application of various drugs, and changes in initial EPSC amplitude and steady-state amplitude upon application of various drugs. For voltage-ramp-evoked currents, data points at at 4-ms intervals were analyzed with a repeated-measures ANOVA. One-way ANOVAs with contrasts were use to assess differences in IPSC kinetics, CPCCOEt-dependent increases in prolonged post-inhibitory rebound firing, spontaneous rates, action potential half-width, normalized protein expression for Western blots, and rotarod Day 1 latency and Δ latency. One-way ANOVAs were used to compare tonic and phasic currents, as well as IPSC kinetics, between control and CPCCOEt-containing solutions across all four groups. Mixed ANOVAs, followed by post-hoc one-way ANOVAs or repeated-measure ANOVAs as appropriate, were used in the rotarod experiments to compare latency to fall across days for all four groups. Two-tailed Student's t-tests were used to compare C57BL/6 males and females. For plotting differences in tonic currents with and without CPCCOEt, the mean (μ) and SEM were calculated from unpaired measurements as μ= $\mu_{control}$-$\mu_{CPCCOEt}$ and SEM = $\sqrt{((SEM_{control})^2-(SEM_{CPCCOEt})^2)}$.

## Acknowledgements

We are grateful to Professor Theo Palmer (Stanford University) for the kind gift of floxed *Gabrb3* mice. We thank members of the Raman lab (Dr. Marion Najac, Dr. Rashmi Sarnaik, Thomas Harmon, Yeechan Wu, and Spencer Brown) for their comments on an earlier version of the manuscript.

## Additional information

### Competing interests
IMR: Reviewing editor, *eLife*. The other authors declare that no competing interests exist.

### Funding

| Funder | Grant reference number | Author |
| --- | --- | --- |
| Simons Foundation | SFARI grant 204813 | Indira M Raman |
| National Institute of Neurological Disorders and Stroke | R37-NS039395 | Indira M Raman |
| National Institute of Mental Health | R01-MH095248 | Catherine S Woolley |

The funders had no role in study design, data collection and interpretation, or the decision to submit the work for publication.

### Author contributions
AAM, Conception, design, analysis, interpretation of all experiments, Data acquisition for all electrophysiological and behavioral experiments, Writing and revising manuscript; KJP, Contribution to data acquisition and analysis for behavioral experiments, Commented on final manuscript; NT, Data acquisition for molecular biological experiments, analysis, interpretation, Commented on final manuscript; CSW, Analysis, interpretation of molecular biological experiments, Commented on final manuscript; IMR, Conception, design, analysis, interpretation of all experiments, Writing and revising manuscript

### Author ORCIDs
Catherine S Woolley, http://orcid.org/0000-0002-8069-2646
Indira M Raman, http://orcid.org/0000-0001-5245-8177

### Ethics
Animal experimentation: All procedures conformed to institutional guidelines and were approved by the Institutional Animal Care and Use Committee of Northwestern University (Animal Welfare Assurance Number, A3283-01; IACUC Study #IS00000242).

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
