## [Decision Letter]

Thank you for sending your work entitled "Sex difference in cerebellar synaptic transmission and sex-specific response to an autism-linked Gabrb3 mutation in mice" for consideration at *eLife*. Your article has been favorably evaluated by Eve Marder (Senior editor), Michael Hausser (Reviewing editor), and two reviewers, one of whom, David Linden, has agreed to reveal his identity.

The Reviewing editor and the other reviewers discussed their comments before we reached this decision, and the Reviewing editor has assembled the following comments to help you prepare a revised submission.

This study presents evidence for sex differences in synaptic physiology and excitability of cerebellar nuclear neurons in weanling mice, which correlate with differential performance on a motor learning task. Overall it was agreed that this work has been carried out to a high standard, and that the results are surprising and potentially of significant interest. However, both reviewers expressed concerns about aspects of the experimental design and about some of conclusions that have been drawn from the results. We are including the detailed comments of the reviewers for your information, and for you to use to guide your revision.

The following issues should be addressed in the revision:

1) Perform new experiments with direct recording of the mGluR1/5 currents (and ideally also experiments with a different antagonist to CPCCOEt). We view these as controls to buttress the conclusions of the manuscript.

2) Strengthen the link between the slice data and the behavioural results (e.g. reporting mouse-by-mouse physiology:behaviour correlations if these data are available).

3) Do some editorial revisions to clarify the conclusions; possibly develop a mechanistic hypothesis for how sex-specific mGluR differences contribute to the observed behavioural differences; importantly address and explain differences with the prior literature. Please step back from the manuscript and help the reader navigate the large number of conditions (sex, developmental time, etc.).

---

## [Author Response]

This study presents evidence for sex differences in synaptic physiology and excitability of cerebellar nuclear neurons in weanling mice, which correlate with differential performance on a motor learning task. Overall it was agreed that this work has been carried out to a high standard, and that the results are surprising and potentially of significant interest. However, both reviewers expressed concerns about aspects of the experimental design and about some of conclusions that have been drawn from the results. We are including the detailed comments of the reviewers for your information, and for you to use to guide your revision.

Thank you for the reviews on our manuscript. We have tried to address both the letter and spirit of the reviews by conducting many additional experiments, further analysing existing data, adding data from C57BL/6 mice to confirm and further test sex differences seen in wild-type *Gabrb3* sibling controls, and rewriting the manuscript substantially. We apologise for the delay in resubmission; our mouse colony collapsed owing to construction outside the animal facility and had to be re-established from cryopreserved mice and mice imported from other universities, which delayed us by many months. To summarise, the new data incorporated into the manuscript include the following:

1) TBOA and antagonist experiments. We recorded the mGluR1/5 currents directly by addition of TBOA. These results opened up an additional line of experiments, which demonstrated the following:

a) Figure 3: With transport blocked, all 4 groups of animals (males/females; wt/mutants) showed very large standing currents that were largely blocked by mGluR1/5 antagonists. Together the results suggest that the basis for the smaller mGluR1/5 currents seen in wild-type males is likely to come from reduced access of glutamate to its receptors, e.g. because of transporters or receptor location, rather than downstream changes or differences in receptor expression (see also points 2 and 3 below).

b) Figure 4: These experiments also allowed a comparison of evoked EPSCs, through AMPARs and mGluR1/5, in all four groups (measured in control, TBOA, and mGluR1/5 blockers). The results permit further specification of sex differences and changes with the mutation.

c) Figure 3 and Figure 4: These experiments also permitted us to test whether the effects of CPCCOEt could be mimicked by the specific mGluR1 antagonist JNJ16259685 combined with the specific mGluR5 antagonist MPEP. In cells tested with the combination of specific antagonists, the results were indistinguishable from data obtained in cells tested with CPCCOEt, lending support to interpretations of the original pharmacological experiments.

2) mGluR1 and mGluR5 expression (Figure 3). We also performed Western blots to test whether the basis for the difference between the sexes and changes with the mutation were likely to result from changes in receptor expression. The data demonstrate that the expression of mGluR1 and mGluR5 are comparable between the sexes and with the mutation, ruling out changes in expression as a primary basis for the differences seen. The most strongly supported alternative remains access of glutamate to receptors (from the TBOA experiments).

3) GABA_A_R expression (Figure 1). We also did Western blots that demonstrated that GABA_A_R β3 in the cerebellum is expressed more heavily in the cortex than the nuclei, that expression is comparable in males and females, and is equivalently reduced in m-/p+ animals. These data also help provide a baseline for more mechanistic interpretations.

4) Further analysis of GABA_A_R kinetics (Figure 1). We also examined the changes in IPSC kinetics over trains of stimulation and calculated the extent to which IPSCs were sufficient to account for changes in tonic current. These data provide a clearer rationale for the investigation of non-GABA_A_R contributions to the differences in synaptic responses.

5) C57BL/6 experiments. Owing to a limited source of mutant mice, we repeated several experiments in C57BL6 mice, both for sex differences in physiology (Figure 1—figure supplement 1), in protein expression, and the mechanism of TBOA action in wild-type animals (Figure 3). These data corroborate the data obtained in male and female mice that were wild-type siblings of the m-/p+ mice, and support the conclusions regarding sex differences.

6) Purkinje-specific m-/p+ experiments (Figure 6 and Figure 7). To strengthen the link between the physiological experiments done in CbN cells and the behavioural results, which admittedly could have a wide range of origins given the global mutation, we repeated both sets of experiments on mice lacking the maternal allele of *Gabrb3* exclusively in Purkinje cells. The results with the Purkinje-specific deletion replicate not only the behavioural differences but also the physiological difference in tonic synaptic current seen with the global mutation. These results provide a stronger framework for interpreting the changes seen in male

*The following issues should be addressed in the revision:*

*1) Perform new experiments with direct recording of the mGluR1/5 currents (and ideally also experiments with a different antagonist to CPCCOEt). We view these as controls to buttress the conclusions of the manuscript.*

We have done the requested experiments. As suggested, we made recordings in control solutions, washed on TBOA, followed by mGluR1/5 antagonists (in the continued presence of TBOA). The results are shown in Figure 3 and Figure 4, and are discussed in the Results section, paragraphs 8-14. Large mGluR1/5-dependent currents are elicited in TBOA. Together the data support the idea that wild-type male mice have smaller evoked mGluR1/5 currents than the other groups. The data also suggest that the basis for this difference is the accessibility of glutamate to mGluR1/5. In these same experiments, we have also used a combination of the selective mGluR1 antagonists JNJ16259685 and mGluR5 antagonist MPEP (n=29), which had the same effect as CPCCOEt (n=10). These results are reported in Figure 3 and Figure 4, and are discussed in the text, Results section paragraph 8 and subsection “Electrophysiological recording”. These results are consistent with a previous comparison of these antagonists that we did previously (Zheng and Raman 2011), which is also now stated explicitly in the manuscript (Results section paragraph 6).

*2) Strengthen the link between the slice data and the behavioural results (e.g. reporting mouse-by-mouse physiology:behaviour correlations if these data are available).*

We addressed this point by repeating both behavioural and electrophysiological experiments in mice with the *Gabrb3* allele deleted from the maternal chromosome in Purkinje cells only. For the rotarod test, 40 total mice were tested. The data show that the Purkinjespecific deletion is sufficient to recapitulate the behavioural differences seen with the global mutation. Moreover, the electrophysiology, done in 32 cells, illustrates that the same changes in tonic current in CbN cells are seen with the Purkinje-specific mutation, and support the idea that the changes seen in mutant male mice are compensatory, in a manner that would offset the expected increase in inhibition to CbN cells owing to the mutation. The data are shown in Figure 6 and Figure 7, and discussed in the text Results section, paragraphs 21-23. These results strengthen the interpretation that the mutation-dependent changes in Purkinje cells trigger alterations in CbN cell physiology, which in turn are linked to the measured behavior.

3) Do some editorial revisions to clarify the conclusions; possibly develop a mechanistic hypothesis for how sex-specific mGluR differences contribute to the observed behavioural differences; importantly address and explain differences with the prior literature. Please step back from the manuscript and help the reader navigate the large number of conditions (sex, developmental time, etc.)

The manuscript has been largely rewritten to clarify the description of experimental results, strengthen the conclusions, and place the work in context of the literature; this has been facilitated by the additional experiments of Figure 3, Figure 4, Figure 6, and 7. In addition, we have measured expression of mGluR1, mGluR5, and GABA_A_R β3, which demonstrate that changes or differences in receptor expression are unlikely to account for the physiological differences observed (Figure 1; Figure 3). We have included a diagram illustrating the proposed compensation for increased inhibition by mGluR1/5 upregulation in males but not females (Figure 8, Discussion section paragraphs 2-3). Please note that the difference from the literature noted by the reviewers does not take into account the developmental changes that we have observed, as it involved comparing weanling animals (this study) with adult animals (the published work of DeLorey et al.). Figure 6—figure supplement 2 includes an illustration of changes in rotarod performance with age which resolves the apparent discrepancies with DeLorey et al., discussed in the Discussion section paragraph 20.